# Spatial control of avidity regulates initiation and progression of selective autophagy

David M. Hollenstein [1,2,3,7], Mariya Licheva [3,4,7], Nicole Konradi[3], David Schweida[1], Hector Mancilla[3], Muriel Mari [5], Fulvio Reggiori [5] & Claudine Kraft [3,6✉]

Autophagosomes form at the endoplasmic reticulum in mammals, and between the vacuole and the endoplasmic reticulum in yeast. However, the roles of these sites and the mechanisms regulating autophagosome formation are incompletely understood. Vac8 is required for autophagy and recruits the Atg1 kinase complex to the vacuole. Here we show that Vac8 acts as a central hub to nucleate the phagophore assembly site at the vacuolar membrane during selective autophagy. Vac8 directly recruits the cargo complex via the Atg11 scaffold. In addition, Vac8 recruits the phosphatidylinositol 3-kinase complex independently of autophagy. Cargo-dependent clustering and Vac8-dependent sequestering of these early autophagy factors, along with local Atg1 activation, promote phagophore assembly site assembly at the vacuole. Importantly, ectopic Vac8 redirects autophagosome formation to the nuclear membrane, indicating that the vacuolar membrane is not specifically required. We propose that multiple avidity-driven interactions drive the initiation and progression of selective autophagy.

[1] Department of Biochemistry and Cell Biology, Max Perutz Labs, University of Vienna, Vienna BioCenter (VBC), Dr. Bohr-Gasse 9, 1030 Vienna, Austria. [2] Vienna BioCenter PhD Program, Doctoral School of the University of Vienna and Medical University of Vienna, Vienna, Austria. [3] Institute of Biochemistry and Molecular Biology, ZBMZ, Faculty of Medicine, University of Freiburg, 79104 Freiburg, Germany. [4] Faculty of Biology, University of Freiburg, 79104 Freiburg, Germany. [5] Department of Biomedical Sciences of Cells & Systems, University of Groningen, University Medical Center Groningen, Groningen, The Netherlands. [6] CIBSS - Centre for Integrative Biological Signalling Studies, University of Freiburg, 79104 Freiburg, Germany. [7]These authors contributed equally: David M. Hollenstein, Mariya Licheva. ✉email: kraft@biochemie.uni-freiburg.de

Macroautophagy, hereafter referred to as autophagy, is an intracellular degradation and recycling pathway that is highly conserved among eukaryotes. During autophagy, cellular material referred to as cargo is engulfed within a newly formed double membrane vesicle, the autophagosome. Once complete, the outer autophagosomal membrane fuses with the lytic compartment (the lysosome in mammals or the vacuole in yeast and plants) releasing the inner vesicle and its cargo for degradation and recycling.

Autophagy is initiated by the formation of a transient structure called the phagophore assembly site (PAS, also known as the pre-autophagosomal structure), which defines the cellular location of autophagosome formation[1,2]. The PAS forms differently depending on the type of autophagy: selective or nonselective. Whereas cytoplasmic material is nonspecifically sequestered by autophagosomes in response to starvation during nonselective or "bulk" autophagy, selective autophagy involves the specific recognition of a cargo by dedicated receptors[3,4]. PAS initiation in bulk autophagy does not require cargo, whereas PAS formation in selective autophagy is triggered by the assembly of the core autophagy machinery on cargo–receptor complexes.

The cytoplasm-to-vacuole targeting (Cvt) pathway is a biosynthetic selective autophagy pathway in S. cerevisiae that delivers aminopeptidase 1 (Ape1) oligomers to the vacuole. The cargo receptor autophagy-related 19 (Atg19) binds to both the Ape1 cargo and the scaffold protein Atg11[5]. Subsequently, the Atg11-bound cargo is recruited to the vacuole, where it colocalizes with the Atg14-containing PI3-kinase complex 1 (PI3KC1) into a distinct structure as part of the PAS[1,6]. Atg11 also recruits the Atg1-kinase complex to the PAS[7–9]. Similarly in mammals, FIP200-bound cargo and PI3KC3–C1 (class-3 PI3KC1) are both recruited to specific ER domains, where they colocalize during PAS and autophagosome formation. FIP200 is also required for recruitment of the ULK1 kinase complex[10–12]. Nevertheless, it remains unknown how Atg11/FIP200-bound cargo and the PI3K complex are recruited to membranes, the vacuole or ER respectively, and whether this specific localization fulfills a functional role during PAS assembly and autophagosome formation.

Vac8 is a vacuole-anchored protein known for its role in vacuole inheritance, homotypic vacuole fusion, and nucleus–vacuole junctions, but has also been found to be required for both selective and bulk autophagy function[13–18]. During bulk autophagy, Vac8 binds Atg13, a subunit of the Atg1 kinase complex[16], which results in the recruitment of other Atg proteins to assemble the bulk PAS at the vacuole[19–22]. Atg13, however, is dispensable for the recruitment of Atg11-bound cargo complexes to the vacuole[8] and thus dispensable for the initiation of selective PAS formation. It therefore remains unclear which role Vac8 plays during selective autophagy.

Detailed insight into the regulation of PAS assembly and the role of membranes has been obscured due to technical limitations. Classical in vitro systems provide the advantage of a chemically well-defined environment but may not sufficiently recapitulate essential cellular contexts, such as distinct protein localizations or membrane compositions[23,24]. On the other hand, in vivo approaches also have limitations, as for instance deletion of a PAS-assembly factor will compromise its earliest role in the assembly process, effectively masking parallel or later requirements. Therefore, the interplay and dependencies of PAS-assembly factors and the importance of their subcellular location have largely remained elusive.

To overcome this limitation, we developed a synthetic reconstitution approach in intact cells that allowed us to systematically dissect the regulatory principles of PAS assembly. Using this system, we discovered that the mechanisms underlying PAS formation at the vacuole are based on avidity, achieved by clustering and compartmentalization. We found that Vac8 serves as a central hub to confine and coordinate the cargo and multiple autophagy factors at the vacuole in parallel, and thus is required to initiate autophagosome formation. Strikingly, we were able to rescue selective autophagy-defective vac8Δ yeast mutants by redirecting PAS assembly to the nuclear membrane with an ectopic Vac8.

## Results

**Vac8 anchors the selective PAS at the vacuole.** Cargo receptors bind to Atg11, which serves as the scaffold to initiate selective autophagy. Atg11 anchors the cargo-receptor complex to the vacuole and recruits other autophagy factors, such as Atg1, to form the PAS[5,25]. How Atg11 is recruited to the vacuole remains unknown. To determine if other Atg proteins are required to anchor Atg11 at the vacuole, we tested seven Atg proteins from the main functional groups: Atg1 and Atg13 (Atg1/ULK1 kinase complex), Atg9 and Atg2 (Atg9 and Atg2–Atg18 system), Atg14 (PI3KC1), Atg12 (Atg12–Atg5–Atg16 conjugation system), and Atg8. We monitored the localization of the fluorescently labeled selective autophagy cargo BFP-Ape1, which was expressed in the respective deletion mutant strains. As expected, deletion of Atg11 abrogated BFP-Ape1 recruitment to the vacuole, detected by staining with FM4–64[5,25]. In contrast, BFP-Ape1 was recruited to the vacuole in atg1Δ, atg13Δ, atg9Δ, atg14Δ, atg12Δ, atg2Δ, and atg8Δ mutants, similar to wild-type cells (Fig. 1a, b and Supplementary Fig. 1a). Thus, canonical Atg proteins are not required to anchor Atg11-bound cargo complexes at the vacuole.

Cargo recognition by autophagy receptor proteins results in local clustering of Atg11 on the cargo complex. To test if clustering of Atg11 is sufficient for its vacuolar localization, we fused Atg11-GFP to a fragment of the reoviral nonstructural protein µNS, which forms multimeric structures of heterogeneous size when expressed in yeast[26] (Fig. 1c and Supplementary Fig. 1b). Both Atg11-GFP-µNS and control GFP-µNS accumulated mostly as a large single cytosolic punctum in atg11Δ atg19Δ and atg11Δ atg13Δ atg19Δ cells, but unlike GFP-µNS, Atg11-GFP-µNS was always associated with the vacuole (Fig. 1d, e). Moreover, we observed a dramatic bending of the vacuolar membrane around Atg11-GFP-µNS, suggesting that multiple interactions between Atg11-GFP-µNS and a vacuolar binding partner deform the vacuolar membrane. Atg11-GFP-µNS particles were not taken up into the vacuolar lumen but remained attached to the outer leaflet of the vacuolar membrane. Importantly, the Atg11-GFP-µNS particles maintained characteristics of endogenous Atg11, such as binding to Atg1, Atg9 and Atg19, as well as its ability to self-interact (Supplementary Fig. 1c, d). Moreover, the interaction of Atg11-GFP-µNS with the vacuole was independent of the cargo–receptor complex (Fig. 1d). Atg11-GFP-µNS therefore serves as an ideal tool to study vacuolar recruitment of Atg11 in vivo.

Given that Vac8 recruits Atg13 to the vacuole to anchor the bulk PAS, we hypothesized that Vac8 might also interact with Atg11 to facilitate selective PAS assembly. To test this hypothesis, we monitored localization of Vac8-mCherry in cells coexpressing Atg11-GFP-µNS, or Atg13-GFP-µNS as a control. Clustered Atg13-GFP-µNS localized to the vacuole and recruited Vac8-mCherry, leading to its depletion from the rest of the vacuolar membrane (Fig. 1f). Intriguingly, Vac8-mCherry was also redistributed to the vacuolar contact site of Atg11-GFP-µNS, suggesting that Atg11-GFP-µNS specifically recruits Vac8 to this location (Fig. 1f). In contrast, the vacuolar V-type proton ATPase Vph1-mCherry was displaced from Atg11-GFP-µNS contact sites (Fig. 1g). These findings suggest that Vac8 interacts with Atg11, recruiting it to the vacuole.

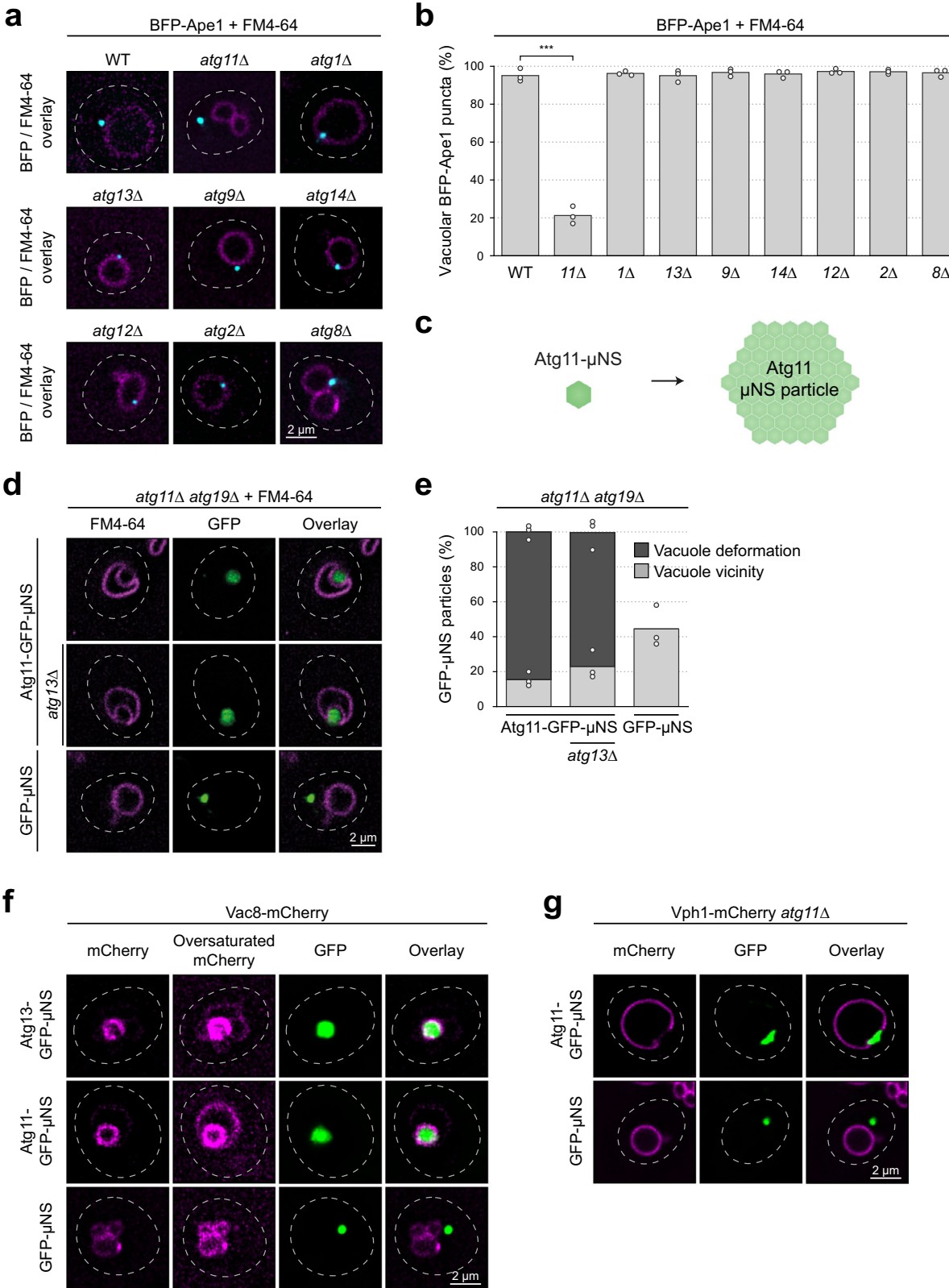

To test if Vac8 is required to anchor the Atg11-cargo complex at the vacuole, we monitored BFP-Ape1 localization in *vac8Δ cells*. In wild-type cells, most BFP-Ape1 puncta were localized in proximity to Vph1-mCherry-containing vacuoles. This localization was largely lost in *atg11Δ* mutants, as expected. *vac8Δ* cells showed vacuolar fragmentation, as previously reported[27]. Importantly, *vac8Δ* cells also displayed a severe defect in vacuole

recruitment of BFP-Ape1 puncta, however less pronounced than in *atg11Δ* cells. Deletion of both, Atg11 and Vac8, did not result in a more severe vacuole-recruitment defect of BFP-Ape1 puncta than a single deletion of *VAC8* (Fig. 2a, b and Supplementary Fig. 2a), indicating that the vacuole fragmentation phenotype of *vac8Δ* cells leads to an increase of false positives in the quantification of vacuole localized puncta, as previously

**Fig. 1 Vac8 clusters at the PAS-vacuole contact site. a** and **b** Wild-type (WT) or indicated *atgΔ* strains containing TagBFP-Ape1 were grown to mid-log phase, and vacuoles were stained with FM4–64. **a** Representative microscopy images showing the overlay of BFP and FM4–64. Dashed lines indicate the contour of individual cells. See also Supplementary Fig. 1a. **b** The number of vacuolar BFP-Ape1 puncta was quantified in three independent experiments. For each strain and replicate at least 120 BFP puncta were analyzed. The values of each replicate (circle) and the mean (bars) were plotted. Statistical analysis using two-tailed unpaired *t*-tests. Significance is indicated with asterisks: \*\*\**p* < 0.001, \*\**p* < 0.01, \**p* < 0.05, n.s. (not significant) *p* > 0.05. Exact numerical values are reported in the source data. **c** Schematic illustration of the Atg11-μNS particle, formed from single monomeric subunits. Note that these particles vary in size and that their structure is unknown. **d** and **e** *atg11Δ atg19Δ* and *atg11Δ atg13Δ atg19Δ* cells containing Atg11-GFP-μNS or GFP-μNS were grown to mid-log phase. Vacuoles were stained with FM4–64. μNS = amino acids 471–721 of the reoviral nonstructural protein μNS. **d** Representative microscopy images are shown. Dashed lines indicate the contour of individual cells. **e** Vacuolar deformation or vacuolar vicinity of Atg11-GFP-μNS and GFP-μNS particles was quantified in three independent experiments. For each strain and replicate at least 70 μNS particles were analyzed. The values of each replicate (circle) and the mean (bars) were plotted. See also Supplementary Fig. 1b. **f** Vac8-mCherry cells expressing Atg13-GFP-μNS, Atg11-GFP-μNS or GFP-μNS were grown to mid-log phase. The intracellular localization of the μNS fusion proteins and the vacuolar distribution of Vac8 was monitored by fluorescence microscopy. Oversaturated images of the mCherry signal allow complete visualization of the vacuolar membrane. Representative fluorescence microscopy images of one out of three independent experiments are shown. Dashed lines indicate the contour of individual cells. **g** Vph1-4xmCherry *atg11Δ* cells containing Atg11-GFP-μNS or GFP-μNS were grown to mid-log phase. Representative fluorescence microscopy images of one out of three independent experiments are shown. Dashed lines indicate the contour of individual cells.

observed[21]. Atg11-GFP-μNS particles also failed to localize to the vacuolar membrane and induce vacuolar membrane bending in *vac8Δ* cells (Supplementary Fig. 2b). To investigate the vacuolar contact of Atg11 at a higher resolution, we analyzed cells expressing Atg11-GFP-μNS or GFP-μNS by transmission electron microscopy (TEM). Clear contacts of Atg11-GFP-μNS spreading along the vacuolar surface were observed in over 80% of cells expressing Atg11-GFP-μNS, whereas such contacts were absent in cells expressing GFP-μNS. These contacts were also absent in *vac8Δ* cells expressing Atg11-GFP-μNS (Fig. 2c, d). Overall, these data demonstrate that Vac8 is required to recruit Atg11 complexes to the vacuolar membrane.

**The interaction of Atg11 with Vac8 is direct and requires clustering.** To decipher which region of Atg11 is required for vacuolar anchoring, we generated a series of Atg11 truncation mutants. We fused these truncations to GFP-Atg19, as they lack the Atg19 binding domain (Fig. 2e[8]). We analyzed GFP localization in *atg11Δ atg19Δ* cells and observed that all truncations formed distinct cytosolic puncta, indicating the efficient recruitment of the fusion proteins to Ape1 oligomers (Supplementary Fig. 3a). As expected, full-length Atg11-GFP-Atg19 puncta localized to the vacuole, marked with Sna3-mCherry, whereas GFP-Atg19 puncta were mostly cytoplasmic without vacuolar contacts (Fig. 2f and Supplementary Fig. 3a). We found that the N-terminal 454 amino acids of Atg11 were necessary and sufficient for vacuolar localization. In addition, we fused Atg11$^{1-454}$ to GFP-μNS and observed vacuolar invaginations and clustering of Vac8 at the vacuole contact site of the particles, similar to full-length Atg11-GFP-μNS (Supplementary Fig. 3b).

To investigate a potential physical association between Vac8 and Atg11, we performed co-immunoprecipitation experiments. We found that full-length Atg11 and Atg11$^{1-454}$ co-immunoprecipitated with Vac8-GFP (Fig. 2g), independently of Atg13 (Supplementary Fig. 3c). In contrast, the known Atg11 interactors Atg1, Atg9, Atg17, Atg19, and Atg29 only co-immunoprecipitated with protein A-tagged full-length Atg11 (PrA-Atg11$^{FL}$), but not PrA-Atg11$^{1-454}$ (Supplementary Fig. 3d). To test if Vac8 and Atg11 interact directly, we performed GST pull-down experiments. Indeed, purified GST-Vac8 bound to His-Atg11 in insect cell lysate, suggesting that this interaction is direct (Fig. 3a). Similarly, Vac8 and Atg11 interacted in yeast-two hybrid analyses, and Atg11$^{1-454}$ was sufficient for this interaction (Supplementary Fig. 4a). The Atg11-Vac8 interaction, however, was very weak compared with the known interaction of Atg11 with Atg19$^{3D}$ (Fig. 3b[28]). Together, these findings demonstrate

that the N terminus of Atg11 interacts directly with Vac8, however, with low affinity.

We hypothesized that if endogenous Vac8 anchors Atg11 at the vacuole, then ectopic localization of Vac8 should anchor Atg11 to another site in the cell. To this end, we employed a tethering system to generate a cytosolic Vac8 site. Briefly, we fused a cytosolic Vac8 truncation mutant to an N-terminal peptide that confers Ape1 binding (ot-Vac8ΔN-BFP; ot = oligomer tether), and expressed this construct in *atg19Δ* cells[5,29]. (Fig. 3c). In this system, Ape1 oligomers are cytosolic and do not interact with other Atg proteins, but induce the clustering of ot-Vac8ΔN-BFP. Therefore, these oligomers can be repurposed to provide a protein-clustering platform that enables the visualization of corecruited proteins by fluorescence microscopy. We recently used this system to show that Atg13 is efficiently recruited to the cytosolic, oligomer-tethered Vac8 site[21].

We confirmed that ot-Vac8ΔN-BFP efficiently recruited Atg13-GFP in *atg11Δ atg19Δ vac8Δ* cells (Fig. 3c, d). Similarly, Vac8ΔN-GFP was efficiently recruited to Atg13-ot-BFP (Supplementary Fig. 4b, c). In contrast, GFP-Atg11 did not interact with ot-Vac8ΔN-BFP (Fig. 3d, e), nor did ot-Atg11-BFP interact with Vac8ΔN-GFP (Supplementary Fig. 4b, c). Similarly, Atg11-GFP did not interact with ot-Vac8ΔN-BFP, however, was efficiently recruited to BFP-Ape1 oligomers (Supplementary Fig. 4d, e). Therefore, in contrast to the Atg13–Vac8 interaction, Atg11 and Vac8 are unable to interact if only one of the binding partners is clustered.

In the wild-type scenario, Atg11 is clustered on the cargo, and Vac8 is spatially confined on the vacuolar membrane, causing a reduced dimensionality of diffusion. The thereby increased effective local concentration of both binding partners could translate the possibly weak individual low affinity interactions into high avidity binding[30]. To test this possibility, we coexpressed ot-Vac8ΔN-GFP with Atg11-BFP-μNS in *atg13Δ atg19Δ vac8Δ* cells, which revealed close association of BFP and GFP puncta (Fig. 3f, g). These findings suggest that the cargo-Atg11 complex is recruited to and anchored at the vacuole by direct interaction of Atg11 with Vac8 in an avidity-driven manner (Fig. 3h).

**Loss of Vac8 prevents PAS maturation during selective autophagy.** Our data establish that Vac8 anchors cargo-clustered Atg11 at the vacuole. The early selective PAS formed by the Atg11-cargo and the Atg1 kinase complex subsequently matures via the recruitment of downstream Atg proteins. First, Atg9-vesicles and Atg14-containing PI3KC1 associate with the PAS, followed by the independent recruitment of the Atg2-Atg18

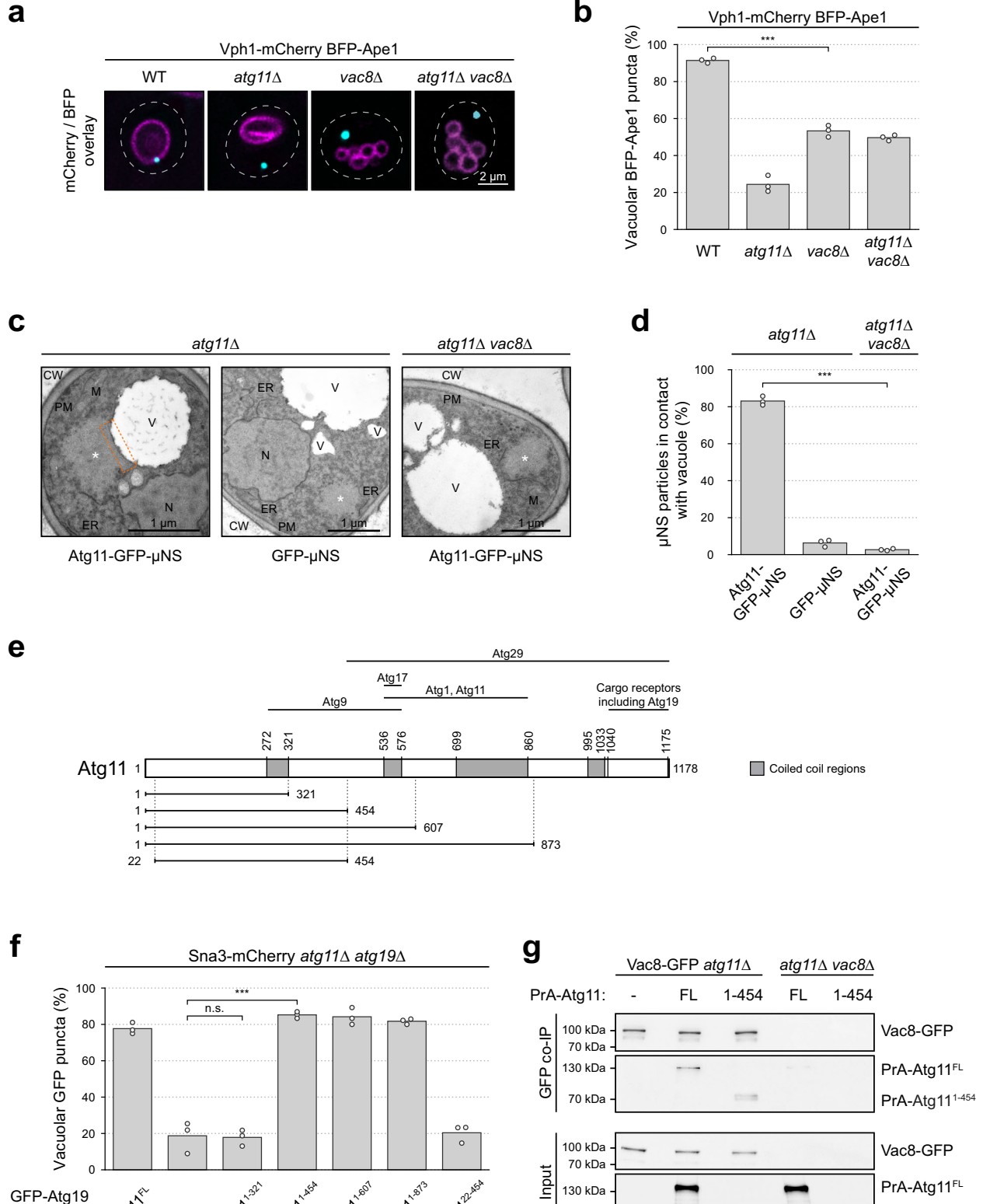

module as well as Atg21, which then results in the recruitment of the Atg12 and Atg8 conjugation systems (Fig. 4a[1]). Whether these downstream events of selective PAS maturation are also affected by Vac8 is unclear.

Under bulk autophagic conditions, the recruitment of the downstream factors Atg2 and Atg8 was not further affected in *vac8Δ* cells. However, *vac8Δ* cells showed an autophagosome–vacuole

fusion defect under starvation conditions. Thus, in bulk autophagy, Vac8 is required for initiation of PAS formation and autophagosome–vacuole fusion, but not for the maturation of the PAS[21].

To test if Vac8 influences PAS maturation in selective autophagy, we monitored the formation of GFP-Atg8 puncta, which represent either the PAS or forming or mature autophagosomes. We observed GFP-Atg8 puncta in over 25% of wild-type

**Fig. 2 Vac8 anchors the selective PAS by interacting with the N-terminus of Atg11. a** and **b** Vph1-4xmCherry wild-type (WT), *atg11Δ*, *vac8Δ*, and *atg11Δ vac8Δ* cells containing TagBFP-Ape1 were grown to mid-log phase. **a** Representative microscopy images showing the overlay of TagBFP-Ape1 and Vph1-4xmCherry signal. Dashed lines indicate the contour of individual cells (see also Supplementary Fig. 2a). **b** The number of vacuolar TagBFP-Ape1 puncta was quantified in three independent experiments. For each strain and replicate at least 27 BFP puncta were analyzed. The values of each replicate (circle) and the mean (bars) were plotted. Statistical analysis using two-tailed unpaired *t*-tests. Significance is indicated with asterisks: ***$p < 0.001$, **$p < 0.01$, *$p < 0.05$, n.s. (not significant) $p > 0.05$. Exact numerical values are reported in the source data. **c** and **d** The indicated strains containing Atg11-GFP-μNS or GFP-μNS were grown to mid-log phase and processed for electron microscopy as described in "Materials and Methods". **c** Representative electron micrographs are shown. CW cell wall, PM plasma membrane, M mitochondria, ER endoplasmic reticulum, N nucleus, V vacuole, asterisk Atg11-GFP-μNS or GFP-μNS particles. The orange dashed box indicates the contact site between Atg11-GFP-μNS and the vacuole. **d** The percentage of μNS particles in contact with the vacuole was quantified in three independent technical replicates. The values of each replicate (circle) and the mean (bars) were plotted. Statistical analysis using two-tailed unpaired *t*-tests. Significance is indicated with asterisks: ***$p < 0.001$, **$p < 0.01$, *$p < 0.05$, n.s. (not significant) $p > 0.05$. Exact numerical values are reported in the source data. **e** Schematic illustration of Atg11 truncations used in this study. Gray boxes indicate previously mapped coiled-coil regions. Known direct interactions are indicated with a black line. **f** Sna3-4xmCherry *atg11Δ atg19Δ* cells expressing the indicated Atg11 truncations fused to GFP-Atg19 or expressing only GFP-Atg19, were grown to mid-log phase. The number of vacuolar GFP puncta was quantified in three independent experiments. For each strain and replicate at least 28 BFP puncta were analyzed. The values of each replicate (circle) and the mean (bars) were plotted. Representative images are shown in Supplementary Fig. 3a. **g** Vac8-GFP *atg11Δ* cells containing 2xProteinA-Atg11^FL, 2xProteinA-Atg11^{1-454} or a control plasmid, or *atg11Δ vac8Δ* control cells expressing 2xProteinA-Atg11^FL or 2xProteinA-Atg11^{1-454} were grown to mid-log phase, followed by glass bead lysis. Vac8-GFP was immunoprecipitated and the amount of precipitated Vac8 and coprecipitated Atg11 was analyzed by anti-GFP and anti-protein A western blotting. One representative experiment out of three independent experiments is shown. FL full length, PrA protein A.

control cells. In contrast, only 8% of *vac8Δ* cells displayed GFP-Atg8 puncta, suggesting that Vac8 is required for PAS maturation and the formation of autophagic membranes (Fig. 4b and Supplementary Fig. 5a). Therefore, Vac8 plays an additional role in selective autophagy that is mediated in a Vac8-independent manner in bulk autophagy.

To test which step during PAS maturation requires Vac8 function, we analyzed the formation of PAS puncta containing Atg14 and Atg9. We used an *atg8Δ* background, which stalls autophagosome formation and prevents PAS turnover. This allows a precise analysis of PAS formation independent of its turnover, which is therefore directly comparable in different mutant situations. The number of cells containing Atg14-GFP puncta drastically dropped from 50% in *atg8Δ* or *atg1Δ atg13Δ atg8Δ* cells to 5% in *atg8Δ atg19Δ* and *atg8Δ atg9Δ* control cells, as expected, given that Atg19 is required to initiate selective autophagy and Atg9 is required to recruit Atg14-containing PI3KC1. Importantly, less than 2% of cells formed Atg14-GFP puncta in *atg8Δ vac8Δ* cells, suggesting that Vac8 is required to recruit Atg14 to the PAS (Fig. 4c, and Supplementary Fig. 5b–d). In contrast, Atg9-GFP vesicles were recruited to BFP-Ape1 puncta with similar frequency in *atg8Δ* and *atg8Δ vac8Δ* cells, but not in *atg8Δ atg19Δ* control cells (Fig. 4d and Supplementary Fig. 5e). These findings suggest that Atg9 vesicle recruitment to the PAS is independent of Vac8, whereas Atg14 recruitment requires Vac8 function.

**Vac8 recruits the PI3KC1 to the vacuole independent of autophagy function.** We noticed that Atg14-GFP localized evenly along the vacuolar membrane in *atg19Δ* cells that lack PAS formation, whereas in *vac8Δ* cells, the vacuolar localization of Atg14-GFP was lost (Fig. 4e and Supplementary Fig. 5f). We therefore speculated that Vac8 is not only required to recruit Atg14 to the PAS, but is also responsible for Atg14 recruitment to the vacuolar membrane in general. To test this possibility, we analyzed Atg14-GFP membrane association in fractionation experiments. Pgk1 and Tom20 served as cytosol and membrane markers, respectively. Whereas Atg14-GFP was found both in the cytosol and membrane fraction of wild-type cells, its membrane association was severely reduced in *vac8Δ* cells. Deletion of *ATG11* did not affect Atg14 levels in the membrane fraction, further supporting that the PAS is dispensable for Atg14 membrane association (Fig. 4f). These results suggest that deletion of

*VAC8* blocks PAS maturation by inhibiting the recruitment of Atg14 to the vacuole.

Atg14 and Atg38 are part of the autophagy specific PI3KC1, whereas Vps30, Vps15 and Vps34 are subunits of both PI3KC1 and the endocytic PI3K complex 2[31]. To test if Vac8 affects vacuolar recruitment of other subunits of the PI3KC1 in addition to Atg14, we analyzed Vps15 localization. Vps15-GFP localized to the vacuolar membrane in wild-type but not in *vac8Δ* cells (Fig. 4g and Supplementary Fig. 5g). These findings suggest that the entire PI3KC1 requires Vac8 for recruitment to the vacuolar membrane.

To test if the PI3KC1 still assembles in the absence of *VAC8*, we performed co-IP experiments. We found that Vps15-GFP and Vps34-GFP co-immunopurified Atg14-TAP in both wild-type and in *vac8Δ* cells. The amount of co-immunopurified Atg14 was even higher in *vac8Δ* cells, suggesting that the PI3KC1 assembles independently of its vacuolar membrane association (Fig. 4h).

If Vac8 directly anchors the PI3K complex at the vacuole, as it does clustered Atg11, then ectopic Vac8 should also recruit PI3K complex members. Indeed, clustered ot-Vac8ΔN-BFP recruited Atg14-GFP in *atg19Δ vac8Δ* cells (Fig. 5a and Supplementary Fig. 6a). To test if the interaction between Atg14 and Vac8 is independent of its interactions with Atg13[16] and Atg11 (Fig. 3a), we employed *atg11Δ atg13Δ atg19Δ vac8Δ* cells coexpressing Atg14-ot-BFP and Vac8ΔN-GFP. We found that Atg14-ot-BFP efficiently recruited Vac8ΔN in both *atg19Δ vac8Δ* and *atg11Δ atg13Δ atg19Δ vac8Δ* cells, suggesting that the interaction of the PI3K complex with Vac8 is independent of Atg11 and Atg13 (Supplementary Fig. 6b, c).

From these results, we hypothesized that Atg14 localizes to the vacuole in a Vac8-dependent, but autophagy independent manner. To test this, we coexpressed Atg14-GFP together with Vps30, which is required to stabilize Atg14[31], in multiple-knockout (MKO) cells that lack 25 Atg proteins[32]. Indeed, Atg14-GFP still localized to the vacuolar membrane in the absence of the core autophagy machinery (Supplementary Fig. 6d).

Moreover, we found that Vps15-GFP was recruited to ot-Vac8ΔN-BFP in both *atg14Δ atg19Δ vac8Δ* and *atg19Δ vac8Δ vps30Δ* cells, suggesting that the interaction of Vac8 and the PI3K complex does not depend on Atg14 and Vps30, and is mediated by Vps15, Vps34 or the Vps15-Vps34 dimer (Fig. 5b and Supplementary Fig. 6e). As *vps34Δ* cells show severely reduced protein levels of Atg14 and Vps15[31], the effect of *VPS34* deletions could not be analyzed in this setup.

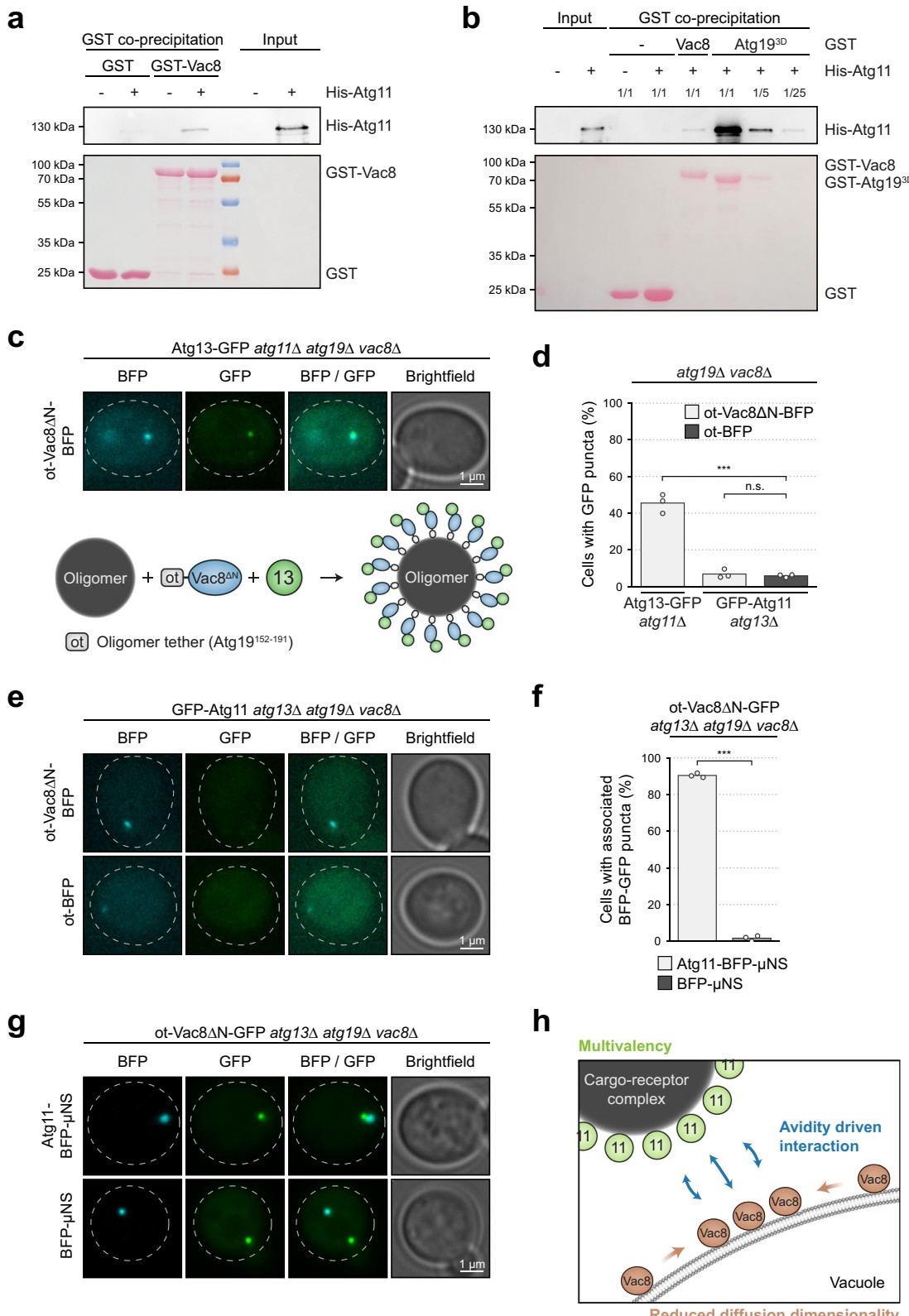

In summary, these findings demonstrate that Vac8 is responsible for the vacuolar recruitment of the PI3K complex via interaction with the Vps15–Vps34 subcomplex. This recruitment does not require Atg14, Atg11, Atg13, nor any other core autophagy protein, thus, PI3KC1 vacuole recruitment is independent of autophagy.

**PI3KC1 PAS association requires vacuolar localization.** Vac8 independently recruits three autophagy modules to the vacuole: the Atg1-kinase complex via Atg13, the PI3KC1, and the Atg11-cargo. If the main function of Vac8 is to tether PAS factors to the vacuole and thereby facilitate PAS maturation, then artificial tethering of these factors to the vacuole should restore PAS assembly in *vac8Δ* cells. To

**Fig. 3 Vac8–Atg11 interaction is direct and requires clustering. a** and **b** GST, GST-Vac8 or GST-Atg19[3D] (S390D, S391D, and S396D) was expressed in *E.coli* and bound to Glutathione Sepharose® (GSH) beads and further incubated with SF9 insect-cell lysates containing overexpressed 8xHis-Atg11 or control lysates. The amount of coprecipitated 8xHis-Atg11 was monitored by anti-Atg11 western blotting, and the amount of GST or GST-Vac8 bound to GSH beads was monitored by Ponceau S staining. One representative experiment out of three independent experiments is shown. GST glutathione S-transferase. **b** A 5 times and 25 times dilution of the GST-Atg19[3D] co-IP was loaded to compare the amount of coprecipitated 8xHis-Atg11 between GST-Vac8 and GST-Atg19[3D]. **c** Atg13-GFP *atg11Δ atg19Δ vac8Δ* cells containing ot-Vac8ΔN-mTagBFP2 (Atg19[152–191]-Vac8[19–578]-mTagBFP2) were grown to mid-log phase. The recruitment of Atg13-GFP to ot-Vac8ΔN-BFP puncta was monitored by fluorescence microscopy. Dashed lines indicate the contour of individual cells. ot oligomer tether. **d** Quantification of **c** and **e**. The percentage of cells with GFP puncta was quantified in three independent biological replicates. For each condition and replicate at least 100 cells were analyzed. The values of each replicate (circle) and the mean (bars) were plotted. Statistical analysis using two-tailed unpaired *t*-tests. Significance is indicated with asterisks: ***$p < 0.001$, **$p < 0.01$, *$p < 0.05$, n.s. (not significant) $p > 0.05$. Exact numerical values are reported in the source data. **e** GFP-Atg11 *atg13Δ atg19Δ vac8Δ* cells containing ot-Vac8ΔN-mTagBFP2 or ot-mTagBFP2 were grown to mid-log phase. The recruitment of GFP-Atg11 to ot-Vac8ΔN-BFP puncta was monitored. Dashed lines indicate the contour of individual cells. **f** and **g** *atg13Δ atg19Δ vac8Δ* cells containing ot-Vac8ΔN-GFP and Atg11-mTagBFP2-µNS or mTagBFP2-µNS were grown to mid-log phase. Colocalization of GFP and BFP puncta was analyzed. **f** The percentage of cells with associated BFP and GFP puncta was quantified in three independent biological replicates. For each condition and replicate at least 35 cells containing BFP and GFP puncta were analyzed. The values of each replicate (circle) and the mean (bars) were plotted. Statistical analysis using two-tailed unpaired *t*-tests. Significance is indicated with asterisks: ***$p < 0.001$, **$p < 0.01$, *$p < 0.05$, n.s. (not significant) $p > 0.05$. Exact numerical values are reported in the source data. **g** Representative fluorescence microscopy images of **f** are shown. Dashed lines indicate the contour of individual cells. **h** Schematic illustration of the Vac8-Atg11 interaction: Atg11 clusters on cargo-receptor complexes, whereas Vac8 is confined at the vacuolar membrane. These local concentrations of Vac8 and Atg11 allow the otherwise low-affinity interactions between Vac8 and Atg11 to be stabilized by avidity.

test this possibility, we tethered Atg11 and Atg14 to the vacuole in *vac8Δ* cells. Specifically, we fused Atg11 to the vacuolar transmembrane protein Vph1 (vt[1]-mScarlet-Atg11, vt[1] = vacuole tether 1) and Atg14 to the transmembrane domain of the vacuolar membrane protein Pho8 (Atg14-GFP-vt[2]).

We found that mScarlet-Atg11 clustered on the cargo but was unable to associate with the vacuole in *vac8Δ* cells, as expected. In contrast, vt[1]-mScarlet-Atg11 displayed stable vacuolar localization in *vac8Δ* cells, which did not require cargo binding via Atg19 (Supplementary Fig. 7a). Similarly, Atg14-GFP-vt[2], but not Atg14-GFP, showed stable vacuole association in the absence of Vac8 (Supplementary Fig. 7b).

To investigate if vacuole tethering of Atg11 and Atg14 restores PAS formation in *vac8Δ* cells, we tethered Atg11, Atg14 or both to vacuoles in BFP-Ape1 containing *vac8Δ* cells. We found that vacuole recruitment of BFP-Ape1 oligomers was restored in *vac8Δ* cells expressing vt[1]-mScarlet-Atg11 (Fig. 5c, column 2). This recruitment was dependent on Atg19, demonstrating that vt[1]-mScarlet-Atg11 recruits Ape1-Atg19 oligomers (Fig. 5c, column 7). However, cytosolic Atg14-GFP was not recruited to puncta containing BFP-Ape1 and vt[1]-mScarlet-Atg11 (Fig. 5c, d, column 2), suggesting that the vacuolar localization of PI3KC1 is required for its association with PAS structures. Intriguingly, Atg14-GFP-vt[2] restored PAS formation of cytosolic mScarlet-Atg11 and BFP-Ape1 to 60% of wild-type levels, suggesting that vacuole-localized PI3KC1 can recruit cargo-Atg11 complexes in the absence of Vac8 (Fig. 5d, column 4). Importantly, coexpression of vt[1]-mScarlet-Atg11 and Atg14-GFP-vt[2] in *vac8Δ* cells restored PAS formation (Fig. 5d, compare columns 1 and 5), which required the Ape1 cargo receptor Atg19 and Atg9 (Fig. 5d, columns 6 and 7), further supporting that Atg9 is required for PI3KC1 to associate with the PAS (Fig. 5c, d). Although Atg14 was dispensable for the initial recruitment of the Atg11-cargo complex to the vacuolar membrane (Fig. 1a), the PI3KC1 might play a role in stabilizing the PAS–vacuole connection during phagophore formation (Fig. 5c, d, column 4).

These findings suggest that vacuolar localization of the PI3KC1 and Atg11-cargo is sufficient to restore their efficient association in the absence of Vac8 (Fig. 5e). In particular, local confinement of PI3KC1 to the vacuole is a prerequisite for its association with the PAS, suggesting that this interaction depends on avidity, similar to Vac8-Atg11. In summary, these findings support that

PAS formation is an avidity-driven process guided by multiple low-affinity interactions, coordinated by Vac8 at the vacuole.

**Reconstitution of PAS formation at the nucleus is sufficient to drive selective autophagy.** To further substantiate that Vac8 acts as a spatial regulator of PAS formation during selective autophagy, we reconstituted this process ectopically in vivo. We used the nuclear membrane as an ectopic site in the cell, as it represents a single and defined membranous structure. In the absence of vacuolar Vac8, such as Vac8ΔN, nucleus–vacuole junctions (NVJs) are lost. Therefore, nuclei do not contact the vacuole in *vac8Δ* mutants or Vac8ΔN mutant cells[14,33,34]. To tether Vac8 to the nucleus, we fused cytosolic Vac8ΔN to the N-terminal, 125 amino acid, transmembrane domain of the nuclear outer membrane protein Nvj1 (nt-mScarlet-Vac8ΔN; nt = nucleus tether). We observed nt-mScarlet-Vac8ΔN distribution along the outer nuclear membrane in *vac8Δ* cells (Fig. 6a).

To test if nt-mScarlet-Vac8ΔN can recruit GFP-Atg11-cargo complexes, we quantified the nuclear association of GFP-Atg11 puncta. Indeed, 60% of GFP-Atg11 puncta were recruited to the nucleus in *vac8Δ* cells expressing nt-mScarlet-Vac8ΔN, but only 10% of GFP-Atg11 puncta were at the nucleus in *vac8Δ* cells expressing nt-mScarlet (Fig. 6b and Supplementary Fig. 8a). Similar to the redistribution of vacuolar Vac8-mCherry to the vacuolar contact sites of Atg11-GFP-µNS (Fig. 1f), also nuclear nt-mScarlet-Vac8ΔN enriched at the nuclear contact sites of Atg11-GFP-µNS. (Supplementary Fig. 8b). These data suggest that nuclear-localized nt-Vac8ΔN efficiently recruits the Atg11-cargo complex.

Since Vac8 recruits PI3KC1 to the vacuole (Fig. 4e, g), we reasoned that nt-mScarlet-Vac8ΔN should recruit PI3KC1 to the nucleus. We monitored Vps15-GFP localization in *vac8Δ* cells expressing nt-mScarlet-Vac8ΔN and found that 60% of cells showed Vps15-GFP recruitment to the nucleus, whereas *vac8Δ* cells expressing nt-mScarlet lacked nuclear localization of Vps15-GFP (Fig. 6c and Supplementary Fig. 8c).

Functional PI3KC1 produces PI3P at the PAS and thereby allows the recruitment of the PI3P-dependent autophagy factors, the Atg2–Atg18 complex and Atg21 (Fig. 4a). To test if PI3P production at this ectopic PAS is functional, we monitored the formation of Atg2-GFP puncta, which is a well-established readout for the production of PI3P at the PAS and the subsequent

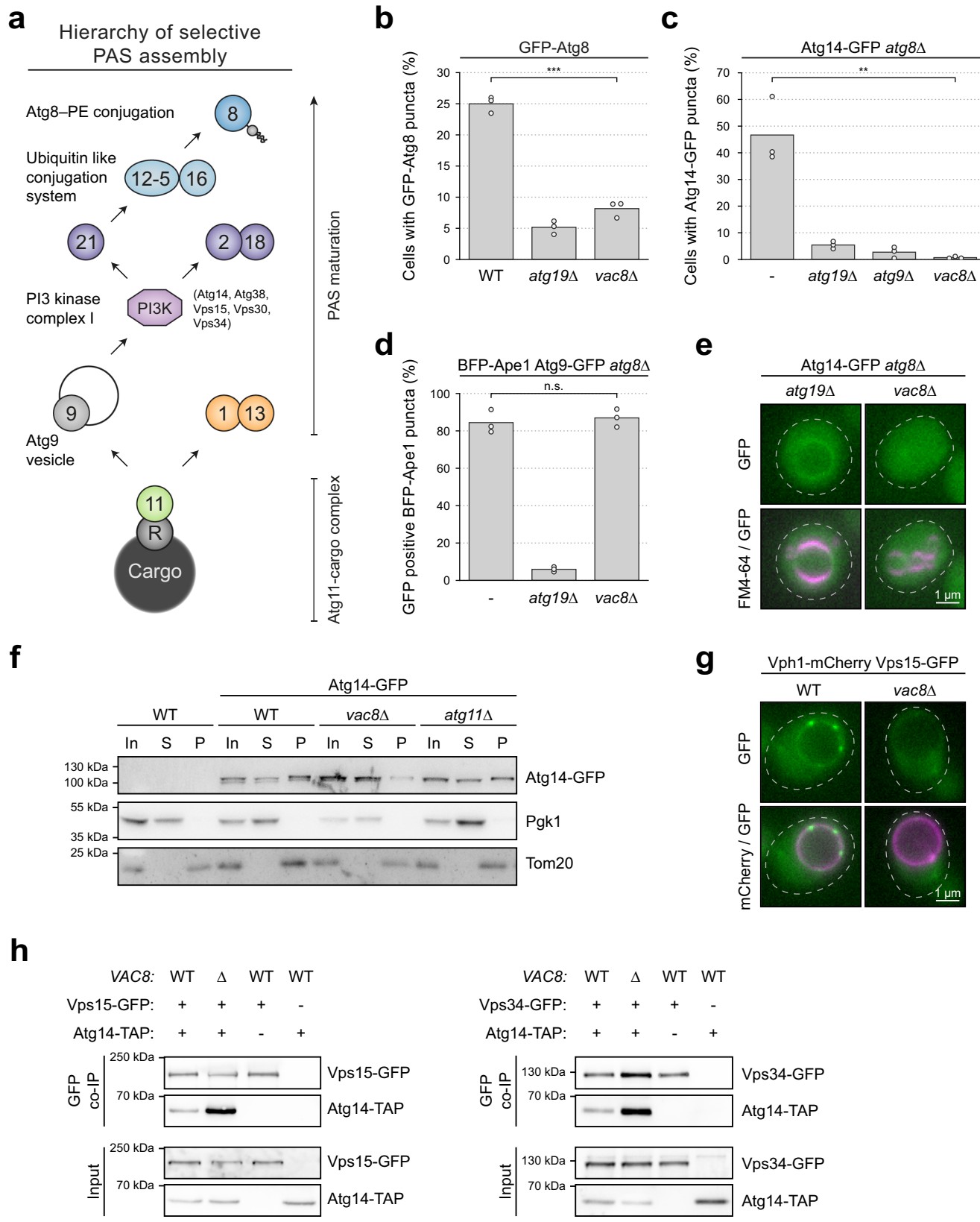

recruitment of the Atg2–Atg18 complex[35,36]. Strikingly, Atg2-GFP formed puncta in *vac8Δ atg8Δ* cells expressing nt-mScarlet-Vac8ΔN, but not in *vac8Δ atg14Δ* cells, *vac8Δ atg11Δ* cells or those expressing nt-mScarlet, suggesting that Atg2-GFP associates with the ectopic PAS in a PI3P-dependent manner (Fig. 6d, and Supplementary Fig. 8d–f).

The final step in PAS maturation is the recruitment of Atg8, which depends on Atg21 and thus PI3P production (Fig. 4a). Similar to Atg2-GFP, GFP-Atg8 puncta also formed in *vac8Δ* cells expressing nt-mScarlet-Vac8ΔN in an Atg14-dependent manner (Fig. 6e and Supplementary Fig. 9a). Thus, ectopic Vac8 stimulates complete PAS maturation at a nonvacuolar membrane.

**Fig. 4 Loss of Vac8 prevents PAS and vacuole recruitment of the PI3K complex. a** Model of hierarchical recruitment of selective PAS factors based on the literature. See also[1] for details. Circles containing numbers correspond to the respective Atg proteins. PE, phosphatidylethanolamine. **b** The indicated strains containing GFP-Atg8 were grown to mid-log phase. The percentage of cells with GFP-Atg8 puncta was analyzed in three independent biological replicates. For each condition and replicate at least 100 cells were analyzed. The values of each replicate (circle) and the mean (bars) were plotted. Statistical analysis using two-tailed unpaired $t$-tests. Significance is indicated with asterisks: ***$p < 0.001$, **$p < 0.01$, *$p < 0.05$, n.s. (not significant) $p > 0.05$. Exact numerical values are reported in the source data. See Supplementary Fig. 5a for representative fluorescence microscopy images. WT wild-type. **c** The indicated strains containing Atg14-3xGFP were grown to mid-log phase. The percentage of cells with Atg14-3xGFP puncta was analyzed in three independent biological replicates. For each condition and replicate at least 100 cells were analyzed. The values of each replicate (circle) and the mean (bars) were plotted. Statistical analysis using two-tailed unpaired $t$-tests. Significance is indicated with asterisks: ***$p < 0.001$, **$p < 0.01$, *$p < 0.05$, n.s. (not significant) $p > 0.05$. Exact numerical values are reported in the source data. See Supplementary Fig. 5b for representative fluorescence microscopy images. **d** The indicated Atg9-GFP strains containing mTagBFP2-Ape1 were grown to mid-log phase. The percentage of GFP positive BFP-Ape1 puncta was analyzed in three independent biological replicates. For each condition and replicate at least 60 BFP puncta were analyzed. The values of each replicate (circle) and the mean (bars) were plotted. Statistical analysis using two-tailed unpaired $t$-tests. Significance is indicated with asterisks: ***$p < 0.001$, **$p < 0.01$, *$p < 0.05$, n.s. (not significant) $p > 0.05$. Exact numerical values are reported in the source data. See Supplementary Fig. 5e for representative fluorescence microscopy images. **e** Vacuolar localization of Atg14-3xGFP was monitored in *atg8Δ atg19Δ* or *atg8Δ vac8Δ* cells. Vacuoles were stained with FM4-64. Cells were grown to mid-log phase. Representative fluorescence microscopy images of one out of two independent experiments are shown. Dashed lines indicate the contour of individual cells (see also Supplementary Fig. 5f). **f** The indicated Atg14-3xGFP strains were subjected to cell lyses and subsequent fractionation. Supernatant and pellet fractions were separated at $20,000 \times g$, and the distribution of Atg14-3xGFP in different fractions was monitored by anti-GFP western blotting. Wild-type cells without GFP-tagged Atg14 were used as a control. Pgk1 was used as a cytosolic marker, Tom20 was used as membrane marker. One representative experiment out of two independent experiments is shown. In, input; S, supernatant; P, pellet fraction. **g** Vacuolar localization of Vps15-GFP was monitored in Vph1-4xmCherry wild-type and *vac8Δ* cells. Cells were grown to mid-log phase. Representative fluorescence microscopy images of one out of two independent experiments are shown. Dashed lines indicate the contour of individual cells (see also Supplementary Fig. 5g). **h** The indicated Vps15-GFP, Vps34-GFP or wild-type strains containing Atg14-TAP or an empty plasmid were grown to mid-log phase. Cell extracts were subjected to GFP immunoprecipitation using GFP-trap affinity resin, and the amount of precipitated protein and coprecipitated Atg14-TAP was analyzed by anti-GFP and anti-protein A western blotting. One representative experiment out of three independent experiments is shown.

Selective autophagosome formation is blocked in *vac8Δ* cells compared with wild-type cells[16]. Our results suggest that nt-mScarlet-Vac8ΔN can completely reconstitute PAS maturation at the nucleus in *vac8Δ* cells. To determine if this ectopic PAS could restore autophagosome formation, we followed GFP-Atg8 puncta formation in the presence or absence of the Rab GTPase *YPT7*. *ypt7Δ* mutants are deficient in autophagosome-vacuole fusion and thus accumulate mature autophagosomes[37]. Indeed, we observed accumulation of GFP-Atg8 puncta in fusion deficient GFP-Atg8 nt-mScarlet-Vac8ΔN *vac8Δ ypt7Δ* cells, suggesting that autophagosomes are formed and fuse with the vacuole in Ypt7 containing GFP-Atg8 nt-mScarlet-Vac8ΔN *vac8Δ* cells (Supplementary Fig. 9b, c). Interestingly, redirecting PAS formation to the nuclear membrane in *vac8Δ YPT7* cells already resulted in an increase of GFP-Atg8 puncta compared with *VAC8* wild-type cells, indicating a reduced fusion efficiency of ectopically formed autophagosomes.

To confirm that autophagosomes are also turned over in these reconstituted cells, we monitored vacuolar uptake of BFP-Ape1. Excitingly, over 40% of *vac8Δ* cells expressing nt-mScarlet-Vac8ΔN showed diffuse vacuolar BFP-Ape1 signal, similar to wild-type cells, and this rescue required the cargo receptor Atg19 (Fig. 7a, b). Vacuolar Ape1 processing was also restored in *vac8Δ* cells expressing nt-mScarlet-Vac8ΔN (Fig. 7c). Thus, reconstitution of an ectopic PAS at the nucleus is sufficient to restore selective autophagy function.

Together, these findings show that Vac8 acts to coordinate PAS formation at the vacuole by recruiting and assembling multiple Atg proteins into the PAS at this membrane site. PAS assembly is governed by avidity, which contributes to the spatiotemporal control of PAS formation during selective autophagy and in succession drives pathway progression (Fig. 7d).

## Discussion

In this study, we developed a synthetic reconstitution approach in live cells to discover the mechanisms underlying PAS assembly. In particular, we dissected the protein interactions and their dependencies that mediate PAS subcellular localization and maturation.

We find that autophagosome formation during selective autophagy in budding yeast is spatially organized by multiple avidity-driven interactions, coordinated at the vacuole by Vac8.

Avidity is an important regulatory feature of many biological systems. The concept of avidity explains how multivalent systems translate individual low affinity interactions into much higher functional affinities. Although each individual interaction may be readily broken, it will likely be restored because the additional interactions prevent structures from diffusing away, thus maintaining a high effective concentration of binding partners. The high avidity of IgM pentamers, for example, renders them particularly efficient at binding antigens present at low levels, but also prevents aberrant interactions[30].

Whereas the concept of avidity has emerged as an important biological feature in extracellular systems, it has been mostly ignored in intracellular regulation[30]. The initiation of selective autophagy represents an intracellular process depending on high avidity interactions. Cargo-dependent clustering of autophagy factors provides multivalency of binding partners. In addition, the restriction of early autophagy factors to a specific membrane site leads to a reduced dimensionality of diffusion and therefore increases their local concentration. These features of PAS formation ensure a robust and decisive assembly process based on binding thresholds, and could make selective autophagy largely irreversible once initiated.

The initiation of bulk autophagy is regulated by the assembly of a higher order oligomeric structure by autophagy scaffold proteins[38]. This assembly provides an increase in the local concentration of early autophagy factors, promoting multivalent interactions, similar to clustering on the surface of selective cargo. It has recently been suggested that this assembly structure also undergoes transient phase separation[19]. In this context phase separation might fulfill an analogous function to clustering on the cargo[39–41], providing avidity by limiting diffusion of phase separated factors.

Vac8 interacts with Atg13, which recruits and confines the Atg1 kinase to the vacuolar membrane. This allows Atg1 to bind Atg11-bound to cargo complexes, resulting in clustering-induced

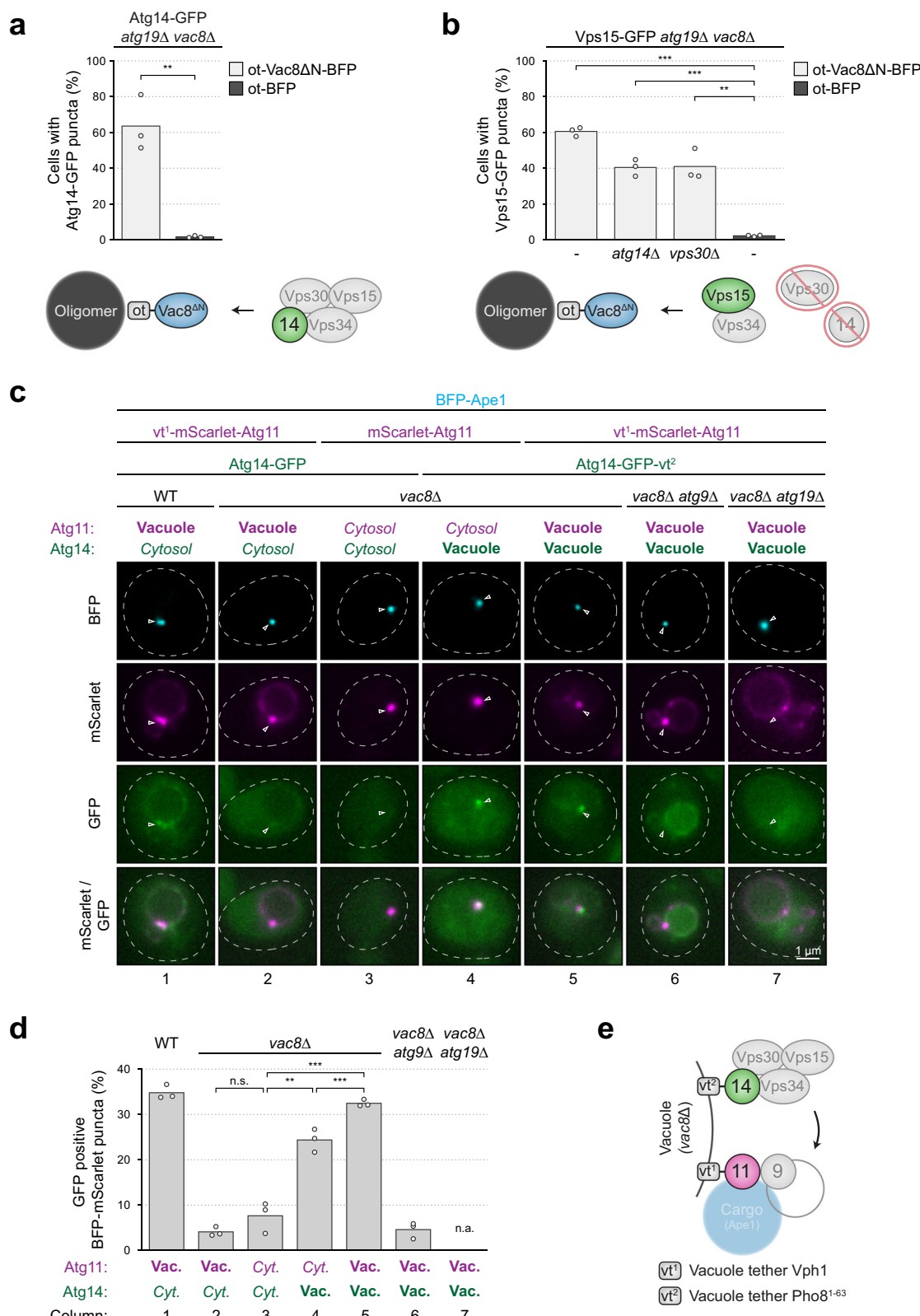

trans-autophosphorylation and local kinase activation at the vacuole[7,8]. Here we find that Vac8 also confines the PI3KC1 to the vacuolar membrane to promote a similar avidity-dependent interaction with the Atg11-cargo complex bound to Atg9 vesicles. Vac8 also directly interacts with Atg11 to recruit the autophagy cargo complex to the vacuole. Stable interaction between Vac8 and Atg11 requires clustering on both sides, demonstrating that

the tethering of the PAS to the vacuole is also achieved via high-avidity interactions. Interestingly, all three Vac8-dependent interactions are independent of each other, suggesting a parallel assembly of the PAS at the vacuolar membrane, rather than a linear hierarchy (Fig. 7d).

This mode of PAS formation displays striking similarities with the assembly of the mammalian autophagy-initiation complex at

**Fig. 5 PI3KC1 PAS association requires vacuole localization. a** Atg14-3xGFP atg19Δ vac8Δ cells containing ot-Vac8ΔN-mTagBFP2 (Atg19[152–191]-Vac8[19–578]-mTagBFP2) or ot-mTagBFP2 were grown to mid-log phase. The recruitment of Atg14-3xGFP to BFP puncta was analyzed by fluorescence microscopy. The percentage of cells with GFP puncta was analyzed in three independent biological replicates. For each condition and replicate at least 100 cells were analyzed. The values of each replicate (circle) and the mean (bars) were plotted. A schematic of the experimental setup is shown below. Statistical analysis using two-tailed unpaired t-tests. Significance is indicated with asterisks: ***$p < 0.001$, **$p < 0.01$, *$p < 0.05$, n.s. (not significant) $p > 0.05$. Exact numerical values are reported in the source data. See Supplementary Fig. 6a for representative fluorescence microscopy images. ot, oligomer tether. **b** The indicated Vps15-GFP strains containing ot-Vac8ΔN-mTagBFP2 or ot-mTagBFP2 were grown to mid-log phase. The recruitment of Vps15-GFP to BFP puncta was analyzed by fluorescence microscopy. The percentage of cells with GFP puncta was analyzed in three independent biological replicates. For each condition and replicate at least 100 cells were analyzed. The values of each replicate (circle) and the mean (bars) were plotted. A schematic of the experimental setup is shown below. Statistical analysis using two-tailed unpaired t-tests. Significance is indicated with asterisks: ***$p < 0.001$, **$p < 0.01$, *$p < 0.05$, n.s. (not significant) $p > 0.05$. Exact numerical values are reported in the source data. See Supplementary Fig. 6e for representative fluorescence microscopy images. **c** and **d** The indicated mTagBFP2-Ape1 strains containing Vph1-mScarlet-Atg11 or mScarlet-Atg11 and Atg14-GFP or Atg14-GFP-Pho8[1–63] were grown to mid-log phase. **c** Representative fluorescence microscopy images are shown. Dashed lines indicate the contour of individual cells, white arrows indicate the position of mScarlet-Atg11 positive mTagBFP2-Ape1 puncta. Note that the BFP and mScarlet channels were individually contrasted due to differences in brightness. **d** The percentage of GFP positive BFP-mScarlet puncta was quantified in three independent biological replicates. For each condition and replicate at least 50 BFP-mScarlet puncta were analyzed. The values of each replicate (circle) and the mean (bars) were plotted. Statistical analysis using two-tailed unpaired t-tests. Significance is indicated with asterisks: ***$p < 0.001$, **$p < 0.01$, *$p < 0.05$, n.s. (not significant) $p > 0.05$. Exact numerical values are reported in the source data. WT, wild-type. **e** Schematic of the experimental setup used in **c** and **d**. vac. vacuole, cyt. cytosol, n.a. not applicable, vt[1] vacuole tether Vph1, vt[2] vacuole tether Pho8[1–63].

the ER membrane. First, FIP200 (Atg11 in yeast) associates with ER membranes, localizing to phosphatidylinositol synthase (PIS)-enriched ER subdomains. Downstream factors are subsequently recruited, initiating phagophore formation at this site[11]. FIP200 and these downstream factors are visible as discrete puncta at the ER in fluorescence microscopy, rather than as a homogeneous reticular ER staining[11,42], indicating that assembly happens at discrete ER subdomains. Similarly in yeast, cargo-bound Atg11 at the vacuolar membrane guides the recruitment of downstream factors to this specific location, leading to the formation of the PAS. In mammals, FIP200 recruits the ULK1 kinase complex, while in yeast Atg11 recruits the Atg1 kinase complex. In both cases, this results in local clustering and activation of the kinase, which is required for autophagy progression. FIP200 associates with the ER independently of PI3KC3–C1, ATG9A and other members of the ULK1 complex. Moreover, PI3KC3–C1 is recruited to the ER likely independently of FIP200 and other autophagy players, whereas FIP200 is required to recruit PI3KC3-C1 to the PIS-enriched ER subdomains[11,43–45]. This is also analogous to Atg11 in yeast, which is recruited to the vacuole independently of the PI3KC1, Atg9 and the Atg1 complex, whereas the PI3KC1-PAS association depends on Atg11. Furthermore, both yeast PI3KC1 and mammalian PI3KC3-C1 require Atg9/ATG9A-positive vesicles to be recruited to the PAS (Fig. 4c[42]). We propose that PAS formation in mammals, like in yeast, involves avidity-dependent interactions that are driven by clustering at the autophagic cargo and confinement on ER membranes. If a central coordinator such as Vac8 also exists in mammals, if this role could be mediated by PIS enrichment, or if independent ER recruitment mechanisms exist for FIP200 and the PI3KC3–C1, remains unknown. During the revision of this paper, ARMC3 has been suggested as a potential homolog of Vac8[46].

Clustered Atg11-μNS induced massive relocalization of Vac8 on the vacuolar membrane to the contact site of Atg11-μNS and exclusion of Vph1 and possibly further vacuolar proteins (Fig. 1f, g[22]). This rearrangement of vacuolar proteins demonstrates their mobility in the 2-dimensional membrane plane and is similar to rearrangements happening at organelle-organelle contact sites, such as Nvj1 enrichment at the nucleus–vacuole junctions or the ERMES (ER–mitochondria encounter structure) at ER-mitochondria contact sites[47–49]. As Vac8 acts as the central hub in organizing the PAS, one can speculate that the mobility of membrane-associated proteins is required for recruiting multiple

complexes into one location at the vacuole. We successfully reconstituted autophagosome formation at a heterologous site, by tethering a cytosolic Vac8 mutant to the outer nuclear membrane. The vacuole per se seems therefore dispensable for this process. If tethering Vac8 to a rigid structure such as a protein oligomer would interfere with its ability to promote PAS formation, and if the fluidity of a membrane is required, will be interesting to address in future studies.

Most autophagy proteins localize to multiple positions at or all along the phagophore during expansion, whereas the PI3KC1 and Atg13 remain only at the junction between the phagophore and the vacuolar membrane[50]. As Vac8 recruits both these factors to the vacuole, Vac8 is likely responsible for their retention at this site. The production of PI3P on the phagophore thus might be spatially restricted to the region on the phagophore that is tethered to the vacuole, from where PI3P is then distributed to the entire phagophore.

In contrast to selective autophagy, deletion of VAC8 did not abrogate Atg2 and Atg8 puncta formation in bulk autophagy[21]. Since the recruitment of both of these proteins depends on PI3P, Vac8 is likely not strictly required for PI3KC1 recruitment to the bulk autophagy PAS. Also, in contrast to selective autophagy, VAC8 deletion cells still form autophagosomes under starvation, however at a substantially reduced rate and smaller in size. This reduced size could stem from a failure in efficient PI3P production at the bulk PAS in vac8Δ cells[21]. How exactly PAS association of the PI3KC1 is regulated in bulk autophagy and if its confinement to the vacuolar membrane is also involved, remains to be determined in more detail. Vac8 in bulk autophagy is furthermore required for the efficient fusion of autophagosomes with the vacuole[21]. Our findings indicate that Vac8 might also be involved in autophagosome–vacuole fusion during selective autophagy (Supplementary Fig. 9b, c).

Clustering of cargo receptors has been recognized as an important regulatory mechanism for selective autophagy. Activation of the Atg1/ULK1 kinase requires their auto-phosphorylation, which is achieved by their clustering on cargo complexes[8,51,52]. The requirement of cargo receptor clustering has also been attributed to phagophore growth, where avidity is required to ensure the stable interaction of cargo complexes with growing autophagic membranes[53–56]. Here clustering of cargo receptors and membrane confinement of Atg8 on the phagophore allow the individual low affinity interactions of receptors with Atg8 to be stabilized by avidity.

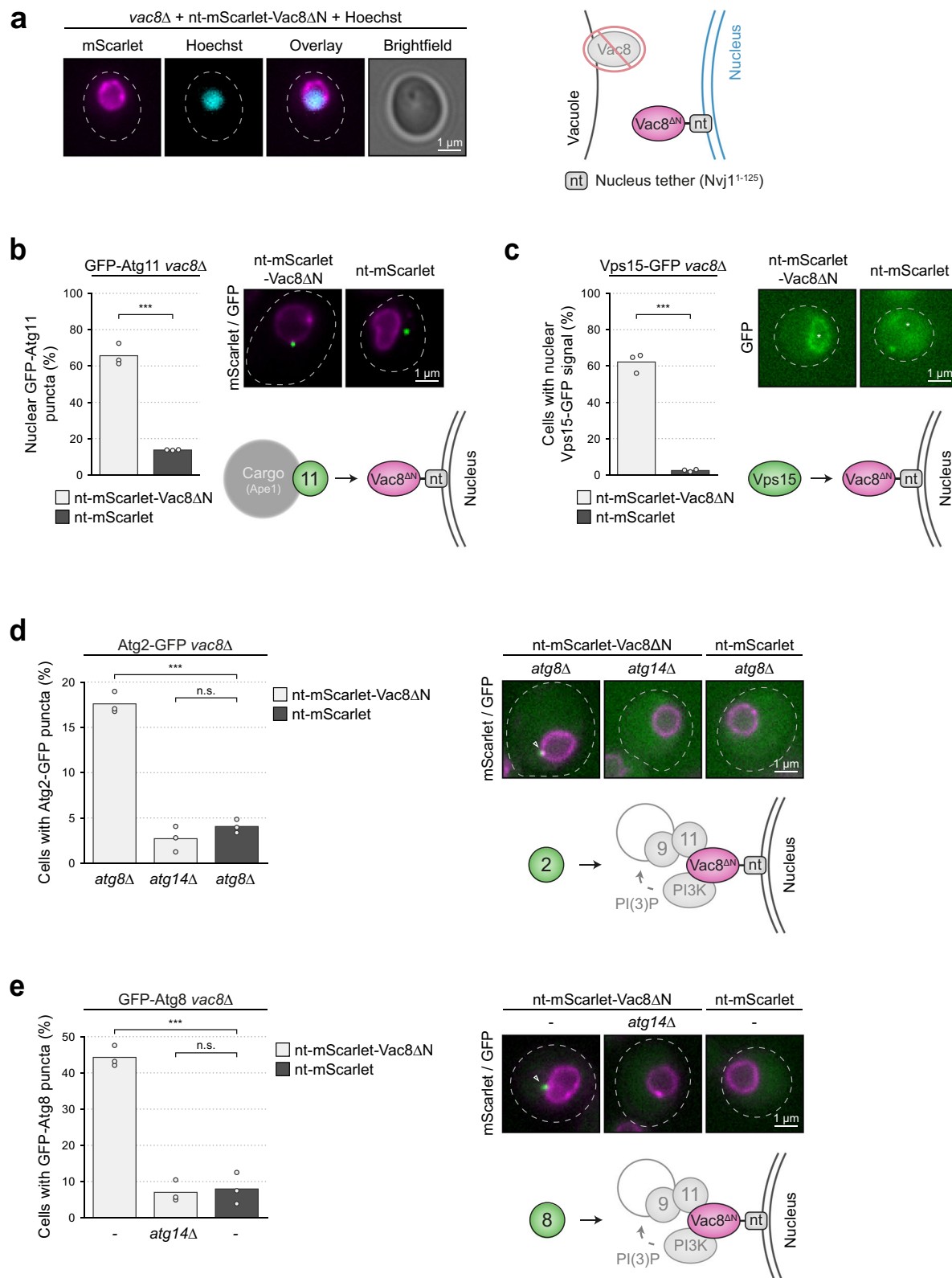

Our findings, however, indicate that this initial cargo receptor clustering not only aids phagophore binding and Atg1/ULK1 kinase activation, but serves to initiate PAS formation by controlling cargo localization and recruitment of the PI3KC1. Following the initial clustering of Atg11 on cargo-receptor complexes and its recruitment to the vacuole, PAS assembly might proceed in a self-organized manner by avidity-driven assembly steps. The recruitment of the Atg machinery in this manner and the many interactions taking place at a confined space support a robust and well controlled assembly process and prevent arbitrary autophagy induction or aberrant autophagosome formation.

**Fig. 6 Reconstitution of PAS formation at the nucleus is sufficient for complete PAS assembly. a** Vph1-GFP *vac8Δ* cells containing nt-mScarlet-Vac8ΔN (Nvj1$^{1–125}$-mScarlet-Vac8$^{19–578}$) were analyzed. The nucleus was stained with Hoechst. Representative fluorescence microscopy images are shown. Dashed lines indicate the contour of individual cells. A schematic of the experimental setup is shown on the right. One representative experiment out of two independent experiments is shown. nt, nucleus tether (Nvj1$^{1–125}$). **b** GFP-Atg11 *vac8Δ* cells containing nt-mScarlet-Vac8ΔN or nt-mScarlet were grown to mid-log phase. The recruitment of GFP-Atg11 puncta to the nuclear membrane was analyzed. The percentage of nuclear GFP puncta was quantified in three independent biological replicates. For each condition and replicate at least 100 GFP puncta were analyzed. The values of each replicate (circle) and the mean (bars) were plotted. Representative fluorescence microscopy images displaying an overlay of the mScarlet and GFP signal and a schematic of the experimental setup are shown on the right. Statistical analysis using two-tailed unpaired *t*-tests. Significance is indicated with asterisks: ***$p < 0.001$, **$p < 0.01$, *$p < 0.05$, n.s. (not significant) $p > 0.05$. Exact numerical values are reported in the source data. See also Supplementary Fig. 8a. **c** Vps15-GFP *vac8Δ* cells containing nt-mScarlet-Vac8ΔN or nt-mScarlet were grown to mid-log phase. The recruitment of Vps15-GFP to the nuclear membrane was analyzed. The percentage of cells with nuclear GFP signal was quantified in three independent biological replicates. For each condition and replicate at least 100 cells were analyzed. The values of each replicate (circle) and the mean (bars) were plotted. Representative fluorescence microscopy images of the GFP signal and a schematic of the experimental setup are shown on the right. White asterisk indicates the position of the nucleus. Statistical analysis using two-tailed unpaired *t*-tests. Significance is indicated with asterisks: ***$p < 0.001$, **$p < 0.01$, *$p < 0.05$, n.s. (not significant) $p > 0.05$. Exact numerical values are reported in the source data. See also Supplementary Fig. 8c. **d** The indicated Atg2-GFP strains containing nt-mScarlet-Vac8ΔN or nt-mScarlet were grown to mid-log phase. The formation of Atg2-GFP puncta was analyzed. The percentage of cells with GFP puncta was quantified in three independent biological replicates. For each condition and replicate at least 100 cells were analyzed. The values of each replicate (circle) and the mean (bars) were plotted. Representative fluorescence microscopy images displaying an overlay of the mScarlet and GFP signal and a schematic of the experimental setup are shown on the right. A white arrow indicates a GFP punctum. Statistical analysis using two-tailed unpaired *t*-tests. Significance is indicated with asterisks: ***$p < 0.001$, **$p < 0.01$, *$p < 0.05$, n.s. (not significant) $p > 0.05$. Exact numerical values are reported in the source data. See also Supplementary Fig. 8d. **e** The indicated GFP-Atg8 strains containing nt-mScarlet-Vac8ΔN or nt-mScarlet were grown to mid-log phase. Experiments were quantified and presented as described in **d**. See also Supplementary Fig. 9a.

## Methods

**Yeast strains and plasmids**. Yeast strains are listed in Supplementary Data 1. Plasmids are listed in Supplementary Data 2, and plasmid sequence maps are available in Supplementary Data 3. Yeast genomic mutations were integrated by homologous recombination; genomic insertions (tagging) were performed according to[57,58] and multiple deletions or mutations were generated by PCR knockout, mating, and dissection. *GFP-ATG8* and *GFP-ATG11* containing strains were generated by crossing with yTB281 and yTB283[8,28], respectively, which had been generated by seamless tagging[59]. yDH459 (*mTagBFP2-APE1*), yDH480 and yDH481 were generated by homologous recombination of *mTagBFP2-APE1:LEU2* from pDH4 into the *leu2Δ0* locus of BY4741. yDH459 was used to cross further strains.

**Growth conditions**. Yeast cells were grown in synthetic medium (SD, 0.17% yeast nitrogen base, 0.5% ammonium sulfate, 2% glucose, and amino acids as required) or rich medium (YPD, 1% yeast extract, 2% peptone, and 2% glucose) to mid-log phase. Yeast cultures were incubated with shaking (220 rpm) at 30 °C.

**Antibodies**. The following primary antibodies were used in this study. Antisera were diluted in 1x PBS pH 7.4 containing 5% milk powder. Mouse monoclonal anti-GFP (1:100, clone 2B6, Merck, cat # MABC1689), rabbit polyclonal PAP antibody (for protein A and TAP detection, 1:3000, Sigma-Aldrich, cat # P1291), mouse monoclonal anti-Pgk1 antibody (1:30,000, clone 22C5D8, Invitrogen, cat # 459250), rabbit polyclonal anti-Atg1 antibody (1:15,000, Daniel Klionsky, University of Michigan, USA), rabbit polyclonal anti-Tom20 antibody (1:500, Nora Voegtle, University of Freiburg, Germany), rabbit polyclonal anti-Atg19 antibody (1:5000, Sascha Martens, University of Vienna, Austria), rabbit polyclonal anti-Ape1 antibody (1:15,000, Claudine Kraft, University of Freiburg, Germany), mouse monoclonal anti-Atg11 antibody (1:500, clone 6F4-G4, Claudine Kraft, University of Freiburg, Germany), mouse monoclonal anti-Atg17 antibody (1:50, clone 4D3-E8, Claudine Kraft, University of Freiburg, Germany), and mouse monoclonal anti-Atg29 antibody (1:25, clone 1C4-D5, Claudine Kraft, University of Freiburg, Germany). The anti-Atg17 and anti-Atg29 antibodies were generated at the Max Perutz Laboratories Monoclonal Antibody Facility by immunizing Balb/c mice with recombinant proteins containing full length Atg17–Atg31–Atg29 purified from Sf9 insect cells. Splenocytes were fused with X63-Ag8.653 myeloma cells, and hybridoma cells were established in HAT selection medium. Hybridoma supernatants were screened for the presence of specific antibodies by immunoblotting, and positive candidates were monoclonalized. The maintenance of mice and the experimental procedures have been conducted according to the Austrian Animal Experiments Act and have been approved by the Austrian Federal Ministry of Science and Research BMWFW-66.009/0211-WF/V3b/2015, and the animal experiments ethics committee of the Medical University of Vienna.

**Standard biochemical assays**. Yeast cell culture or yeast extracts and fractions thereof were precipitated with 7% trichloroacetic acid (TCA) for 30 min on ice or overnight at −20 °C. Precipitated proteins were pelleted at 16,000 × g for 15 min at 4 °C, washed with 1 ml of acetone, air-dried, resuspended in urea loading buffer (120 mM Tris-HCl pH 6.8, 5% glycerol, 8 M urea, 143 mM β-mercaptoethanol, and 8% SDS), boiled, and analyzed by SDS-PAGE. Protein extracts were transferred to nitrocellulose membranes, and proteins were detected by immunoblotting, using the ECL detection system with HRP coupled secondary antibodies. Uncropped western blots are provided in the source data.

**Yeast-extract preparation and immunoprecipitation**. For preparation of yeast extract by freezer milling, cells were harvested by filtration on a 90 mm glass filter (SterliTech) using a nitrocellulose membrane with a pore size of 0.45 μm followed by freezing in liquid nitrogen. Cells were milled in a cryogenic grinder (SPEX Freezer Mill 6875, SPEX SamplePrep), using five rounds of 3 min breakage at 15 cycles per second and 2 min of cooling, and the powder was stored at −80 °C. Freezer milled yeast powder was resuspended in RLB + buffer (1x PBS pH 7.4, 10% glycerol, 0.5% Tween-20, 1 mM sodium fluoride, 20 mM β-glycerol, 1 mM PMSF, 1 mM sodium vanadate and protease inhibitor cocktail [Roche]) and the cell extract was cleared by centrifugation twice at 5000× for 10 min at 4 °C and the supernatant was transferred to a new microfuge tube each time (Supplementary Fig. 3d).

For preparation of yeast extract by glass bead beating, cells were harvested by centrifugation at 3000 × g for 5 min at RT, and resuspended in 1x PBS, pH 7.4, 2% glucose. Cells were pelleted at 3000 × g for 5 min at RT, and resuspended in RLB + buffer. Cells were lysed by the addition of glass beads and vortexing for 6 min at 4 °C. The cell extract was cleared by centrifugation at 573 × g for 5 min at 4 °C (Figs. 2g, 4h, and Supplementary Fig. 3c).

For immunoprecipitation, the protein concentration of cell extracts was adjusted to 20 μg/μl in RLB + buffer. For protein A immunoprecipitation, magnetic Dynabeads$^{TM}$ M-270 Epoxy (Invitrogen) were coupled with IgG from rabbit serum (Sigma). For GFP immunoprecipitation magnetic GFP-Trap® Dynabeads$^{TM}$ (Chromotek) were used. The magnetic beads were incubated with cell extract for 1 h rotating at 4 °C and washed three times with RLB + buffer. The magnetic beads were resuspended in urea loading buffer, boiled and analyzed by SDS-PAGE and western blotting.

**Cell fractionation**. In total, 50 OD$_{600}$ units (one OD$_{600}$ unit corresponds to 1 ml of yeast culture with an OD$_{600}$ of 1) of yeast culture were pelleted at 3000 × g for 5 min at RT, resuspended in DTT buffer (100 mM Tris-HCl pH 9.4 and 10 mM DTT), and incubated for 15 min at 30 °C. Cells were pelleted at 3000 × g for 5 min at RT, resuspended in SP buffer (0.25% YPD, 1 M sorbitol, 50 mM potassium phosphate pH 7.5, and 1 mM DTT), and spheroplasted by lyticase (100 U/ml) treatment for 30 min at 30 °C. Spheroplasts were pelleted at 1500 × g for 5 min at 4 °C and washed with PS1000 buffer (1 M sorbitol, 10 mM PIPES-KOH pH 6.8, and complete$^{TM}$ protease inhibitor cocktail [Roche]). Spheroplasts were pelleted at 1500 × g for 5 min at 4 °C, resuspended in PS200 buffer (200 mM sorbitol, 10 mM PIPES-KOH pH6.8, and complete$^{TM}$ protease inhibitor cocktail [Roche]) and incubated for 15 min at 4 °C. Spheroplasts were lysed on ice by pressing through a 22 G needle 15 times. The lysate was cleared by centrifugation at 500 × g for 5 min at 4 °C and transfer of the supernatant to a new microfuge tube until no pellet was observed. To separate the cytosolic from the membrane fraction, the cleared lysate was spun at 20,000 × g for 20 min at 4 °C. The supernatant was transferred in a new Eppendorf tube and the pellet was resuspended in PS200 buffer. The supernatant

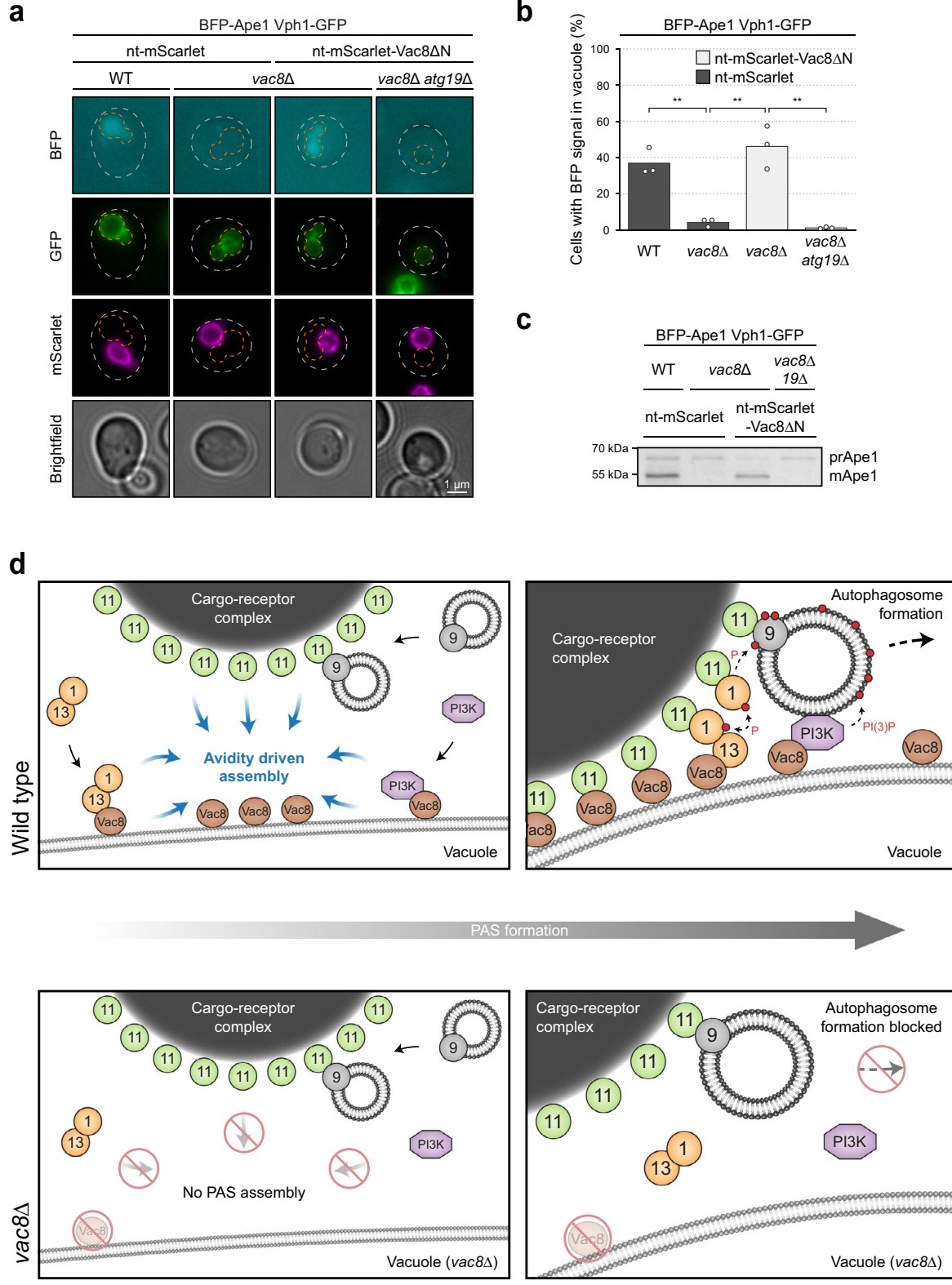

was cleared by centrifugation at 20,000 × g at 4 °C for 20 min and transferred to a new microfuge tube. The pellet fraction was spun at 20,000 × g at 4 °C for 20 min and the pellet was resuspended in PS200 buffer. Cleared supernatant and pellet fractions were TCA precipitated and analyzed by SDS-PAGE and western blotting.

**Yeast two-hybrid (Y2H) assay**. For Y2H experiments Atg11[FL]-GAD or Atg11[1–454]-GAD and LexA-Vac8 fusion constructs were expressed in the *S.cerevisiae* L40 Y2H reporter strain. In total, 5 OD600 units of yeast culture were pelleted at 3000 × g for 5 min at RT, resuspended in 30 μl of 1x PBS pH 7.4 and frozen in

liquid nitrogen. Cells were thawed at RT and mixed with 100 μl of reaction buffer (2% 5-bromo-chloro-3-indolyl-β-D-galactosidase in dimethylformamide, 10% SDS, 1x PBS pH 7.4 and 0.05% β-mercaptoethanol), transferred to 96-well plates, and incubated in the dark at RT. Pictures were taken at several time points within 24 h.

**Protein expression and in vitro-binding assay**. 8xHIS-Atg11 was expressed from baculovirus-infected Sf9 *Spodoptera frugiperda* insect cells (Expression Systems, cat # 94-001 F). The Atg11 ORF was and subcloned with an N-terminal 8xHIS tag into a pLIB library vector[60]. The recombinant bacmid carrying 8xHIS-Atg11 was

**Fig. 7 Vac8 coordinates PAS maturation and defines the site of autophagosome formation. a, b,** and **c** The indicated Vph1-GFP mTagBFP2-Ape1 strains containing nt-mScarlet-Vac8ΔN or nt-mScarlet were grown to mid-log phase. nt, nucleus tether (Nvj1[1–125]). **a** The transport of mTagBFP2-Ape1 to the vacuole was analyzed. Representative microscopy images are shown. Dashed gray lines indicate the contour of individual cells, dashed orange lines indicate the vacuole. Note that the GFP and mScarlet channels were individually contrasted due to differences in brightness. **b** The percentage of cells with vacuolar BFP signal was quantified in three independent biological replicates. For each condition and replicate at least 100 cells were analyzed. The values of each replicate (circle) and the mean (bars) were plotted. Statistical analysis using two-tailed unpaired $t$-tests. Significance is indicated with asterisks: \*\*\*$p < 0.001$, \*\*$p < 0.01$, \*$p < 0.05$, n.s. (not significant) $p > 0.05$. Exact numerical values are reported in the source data. **c** Cell extracts were prepared by TCA precipitation and Ape1 processing was monitored by anti-Ape1 western blotting. WT wild-type, prApe1 precursor form of Ape1, mApe1 mature form of Ape1. **d** Vac8 acts as a vacuolar assembly hub by recruiting Atg11-bound cargo-receptor complexes, the Atg1 kinase complex and the PI3KC1 to the vacuole, facilitating their interaction and PAS formation at this site. Stable interaction between Vac8 and Atg11 depends on avidity, achieved by the local concentration of Atg11 on the cargo and the confinement of Vac8 at the vacuole. Independent of the cargo the Atg1 kinase complex and PI3KC1 are also recruited to the vacuole by Vac8. Assembly of these three Vac8-interacting complexes into the PAS is then further facilitated by avidity driven interactions: The confinement of the Atg1 kinase complex on the vacuolar membrane allows its stable binding to Atg11-bound cargo, whereas vacuolar confinement of Atg14 facilitates its interaction with Atg9 vesicles that are clustered on Atg11-bound cargo. Atg1 recruitment to cargo-bound Atg11 results in clustering-induced Atg1 trans-autophosphorylation and kinase activation at the PAS. Atg9 phosphorylation by Atg1 and PI3P production by the PI3KC1 at this site then promote PAS maturation and ultimately the formation of an autophagosome. In the absence of Vac8, the PAS cannot assemble and autophagosomes do not form. Circles containing numbers correspond to the respective Atg proteins. PI3K PI3 kinase complex I, P protein phosphorylation, PI(3)P phosphatidylinositol 3-phosphate.

assembled in DH10EMBacY E. coli strain (Geneva Biotech). For 8xHIS-Atg11 expression Sf9 cells were grown in 1 l of ESF 921 Insect Cell Culture Medium (Expression Systems) supplemented with penicillin and streptomycin at 27 °C to $1 \times 10^6$ cells/ml, infected by the addition of 1 ml of V1 virus, and grown for 4 days at 27 °C. Cells were pelleted at $500 \times g$ for 10 min at RT, washed with 1x PBS pH 7.4, and the pellet was frozen in liquid nitrogen and stored at −80 °C. Cell pellets were resuspended in lysis buffer (500 mM Tris pH 7.4, 1.5 M KCl, 50 mM MgCl₂, 10% glycerol, 5 mM β-mercaptoethanol, 1 mM PMSF, 0.1% Triton X-100, 10 mM imidazole, complete™ protease inhibitor cocktail [Roche], Benzonase® (25 U/ml; MERCK) was added, and cells were lysed by 12 passes using a Dounce homogenizer (WHEATON® Dounce Tissue Grinder). Two freeze–thaw cycles were performed. Cell lysates were frozen in liquid nitrogen followed by thawing at 4 °C, with additional Benzonase® (25 U/ml, MERCK) being added before each freezing step. Cell lysates were cleared three times by centrifugation at $30,000 \times g$ for 10 min at 4 °C and the supernatant was transferred to a new microfuge tube each time.

GST, GST-Vac8, and GST-Atg19[3D] fusion constructs were expressed from pGEX-4T-1 in E. coli BL21(DE3). Cells were grown in lysogeny broth (LB) medium supplemented with ampicillin at 37 °C, until an $OD_{600}$ of 0.7, the temperature was reduced to 16 °C and expression was induced by addition of 1 mM IPTG for 18 h. Cells were pelleted at $3000 \times g$ for 15 min at RT, resuspended in GST-lysis buffer (50 mM Tris-HCl pH 7.5, 150 mM NaCl, 5% glycerol, 1% Triton X-100, 1 mM PMSF, 1 mM DTT, and complete™ protease inhibitor cocktail [Roche]), and lysed by sonication on ice. Cell lysates were cleared by centrifugation at $16,000 \times g$ for 10 min at 4 °C. The supernatant was incubated with Glutathione (GSH) Sepharose® 4B beads (GE Healthcare) for 1 h rotating at 4 °C. The beads were washed three times by pelleting at $300 \times g$ for 30 sec at 4 °C and resuspension in GST-wash buffer I (50 mM Tris-HCl pH 7.5, 150 mM NaCl, 5% glycerol, 1% Triton X-100, 1 mM DTT and complete™ protease inhibitor cocktail [Roche]), and once with 1x PBS pH 7.4. The beads were incubated with cleared Sf9 insect cell lysate from 8xHIS-Atg11 expressing cells or noninfected control cells, and incubated for 1 h rotating at 4 °C. The beads were washed three times with GST-wash buffer II (50 mM Tris-HCl pH 7.5, 150 mM NaCl, 5% glycerol, 0.1% Triton X-100, 0.5% Tween-20, and 1 mM DTT and complete™ protease inhibitor cocktail [Roche]), and once with GST-wash buffer III (1x PBS, pH 7.4, 10% glycerol, 0.5% Tween-20, 1 mM DTT). The beads were resuspended in urea-loading buffer, boiled and analyzed by SDS-PAGE and western blotting.

**Live-cell imaging.** Exponentially growing cells were placed on 35 mm glass bottom dishes (D35-20 1.5-N, In Vitro Scientific) pretreated with concanavalin A type IV (1 mg/ml, Sigma-Aldrich), and live-cell imaging was performed at RT.

Fluorescent microscopy images were recorded with a DeltaVision Ultra High Resolution microscope (GE Healthcare, Applied Precision) equipped with an UPlanSApo 100x/1.4 oil objective (Olympus), an sCMOS pco.edge camera (PCO), and a seven channel solid state light source (Lumencor) (Figs. 1a, b, d–g, 2a, b, and Supplementary Figs. 1a–d, 2a, b, 3b, 4d, e, 5c, d, 6d, 8b, e, f, 9b, c); or with a PersonalDeltaVision microscope (GE Healthcare, Applied Precision) equipped with an UPlanSApo 100× oil/1.4 oil objective (Olympus), a CoolSNAP HQ2 Monochrome CCD camera (Photometrics), and a seven-color InsightSSI solid state illumination unit (GE Healthcare, Applied Precision) (Fig. 2f and Supplementary Fig. 3a); or with an AxioObserver Z1 inverted microscope (ZEISS) equipped with an EC Plan-Neofluar 100x/1.3 oil M27 objective (ZEISS), a CoolSnap HQ2 Monochrome CCD camera (Photometrics), and a SOLA 6-LCR-SB light source (Lumencor) with the VisiView software (Visitron Systems) (Figs. 3c–g, 4b–e, g, 5a–d, 6a–c, 7a, b, and Supplementary Figs. 4b, c, 5a, b, e–g, 6a–c, e, 7a, b, 8a, c, d, 9a).

Raw microscopy images acquired with the PersonalDeltaVision microscope or the DeltaVision Ultra High Resolution microscope were deconvolved using the softWoRx deconvolution plugin (version R6.1.1 and version 7.2.1, respectively). Image analysis was performed using FIJI[61]. Images from each figure panel were taken with the same imaging setup and are shown with the same contrast settings, unless stated otherwise. Single focal planes of representative images are shown. For quantification three independent replicates were analyzed and manual counting was performed blindly after randomizing image names.

Subcellular positioning of mTagBFP-Ape1 or GFP-μNS particles was investigated by analyzing their localization in regard to the vacuole, by staining vacuoles with the FM™ 4–64 (Thermo Fisher Scientific) dye or by using the genomically tagged vacuole marker protein Vac8-mCherry, Vph1-4xmCherry or Sna3–4xmCherry as indicated. Images were generated by collecting a z-stack of 21 pictures with focal planes 0.25 μm apart or 30 pictures with focal planes 0.20 μm apart. (Figs. 1a, b, d–g, 2a, b, f, and Supplementary Figs. 1a, b, 2a, b, 3a, b).

Size quantification of μNS particles was performed in FIJI on a maximum-intensity z-projection. To generate binarized masks, the same intensity threshold was applied to all images, which were then subjected to particle analysis, using a minimal size of 8 pixels and a circularity between 0.5 and 1. To extract particle sizes, the masks were applied on the z-projected images (Fig. 1d, Supplementary Fig. 1b).

Recruitment of GFP tagged prey proteins to oligomer-tethered bait proteins was analyzed by quantifying the number of cells with GFP puncta, and association of oligomer-tethered ot-Vac8ΔN-GFP with Atg11-BFP-μNS particles was analyzed by counting the number of BFP-puncta overlapping with GFP puncta. Images were generated by collecting a z-stack of 11 pictures with focal planes 0.25 μm apart (Figs. 3c–g, 5a, b, and Supplementary Figs. 4b–e, 6a–c, e).

Quantitative analysis of PASs, forming autophagosomes or completed autophagosomes was performed by counting the number of cells with GFP-Atg8 or Atg14-3xGFP puncta, or by counting the number of mTagBFP2-Ape1 puncta overlapping with Atg9-GFP puncta. Images were generated by collecting a z-stack of 21 pictures with focal planes 0.25 μm apart (Fig. 4b–d, and Supplementary Fig. 5a–e).

Vacuole association of Atg14-3xGFP or Vps15-GFP was investigated by staining vacuoles with the FM™ 4–64 (Thermo Fisher Scientific) dye or by using the genomically tagged vacuole marker protein Vph1-4xmCherry. Images were generated on one focal plane (Fig. 4e, g, and Supplementary Figs. 5f, g, 6d).

Vacuole association of artificially vacuole tethered mScarlet-Atg11 or Atg14-GFP was investigated by using the genomically tagged vacuole marker protein Vph1-GFP or Vph1-4xmCherry, respectively. Images were generated by collecting a z-stack of 21 pictures with focal planes 0.25 μm apart (Supplementary Fig. 7a, b).

Association of Atg14-GFP with mScarlet-Atg11 clustered on mTagBFP2-Ape1 oligomers was analyzed by counting the number of mScarlet-mTagBFP2 double-positive puncta overlapping with GFP puncta. Images were generated by collecting a z-stack of 11 pictures with focal planes 0.25 μm apart (Fig. 5c, d).

Nucleus association of artificially nucleus-tethered nt-mScarlet-Vac8ΔN was investigated by staining of the nucleus with the Hoechst 33258 (Sigma) dye. Images were generated by collecting a z-stack of 11 pictures with focal planes 0.25 μm apart (Fig. 6a).

Subcellular positioning of GFP-Atg11 puncta or Vps15-GFP signal was investigated by analyzing their localization in regard to the nuclear membrane, marked by expression of nucleus tethered nt-mScarlet-Vac8ΔN or nt-mScarlet. Cells not containing nuclear mScarlet signal were excluded from the analysis. Images were generated by collecting a z-stack of 11 pictures with focal planes 0.25 μm apart (Fig. 6b, c, and Supplementary Fig. S8a, c).

Quantitative analysis of ectopic PASs, forming autophagosomes or completed autophagosomes was performed by counting the number of cells with GFP-Atg8, Atg2-GFP puncta, or by counting the number of mTagBFP2-Ape1 puncta overlapping with Atg2-GFP puncta. Cells not containing nuclear mScarlet signal were excluded from the analysis. Images were generated by collecting a z-stack of 11 pictures with focal planes 0.25 μm apart (Fig. 6d, e, Supplementary and Figs. 8d–f, 9a–c).

Transport of mTagBFP2-Ape1 into the vacuole was analyzed by using the genomically tagged vacuole marker protein Vph1-GFP and counting the number of cells containing BFP signal within the vacuolar lumen. Cells not containing nuclear mScarlet signal were excluded from the analysis. Images were generated on one focal plane, except for the BFP channel, 7 pictures with focal planes 0.25 μm apart were collected and the intensity averaged. (Fig. 7a, b).

**Electron microscopy**. Fifteen $OD_{600}$ units of cells were harvested by centrifugation ($1800 \times g$, 5 min, RT). Cells were washed in distilled $H_2O$ and pelleted by centrifugation ($1800 \times g$, 5 min, RT). Cells were resuspended in 3 ml of freshly prepared ice-cold 1.5% $KMnO_4$ (Sigma) and transferred into two 1.5 ml microfuge tubes. After topping up the tube with the same solution to exclude air, samples were mixed for 30 min rotating at 4 °C. After centrifugation ($1400 \times g$, 3 min, 4 °C), the 1.5% $KMnO_4$ incubation was repeated once more before washing the pellets five times with 1 ml of $H_2O$. Permanganate-fixed cells were dehydrated stepwise with increasing concentrations of acetone (10%, 30%, 50%, 70%, 90%, 95% and three times 100%). Each incubation step was performed for 20 min rotating at RT, in-between each step cells were pelleted by centrifugation ($1400 \times g$, 3 min, RT). Cell pellets were resuspended in 33% Spurr's resin in acetone and mixed for 1 h rotating at RT. Cells were pelleted ($7600 \times g$, 3 min, RT) and incubated in 100% freshly made Spurr's resin overnight rotating at RT. This operation was repeated the following day, over the day, after centrifugation of the overnight incubation ($9000 \times g$, 5 min, RT). The Spurr's resin mixture was prepared by mixing 10 g of 4-vinylcyclohexene dioxide (or ERL4206), 4 g of epichlorohydrin-polyglycol epoxy (DER) resin 736, 26 g of (2-nonen-1-yl)succinic anhydride (NSA), and 0.4 g of N,N-diethylethanolamine (all from Sigma). Afterward, the cell and Spurr's mixture was transferred to size 00 embedding capsules (Electron Microscopy Science), and cells were pelleted by centrifugation ($9000 \times g$, 5 min, RT). Embedding capsules were topped up with 100% Spurr's and baked for a minimum of 3 days at 60 °C.

Thin sections of ~55 nm in thickness were cut using an ultramicrotome (Leica Microsystems). Sections were collected on formvar carbon-coated 50-mesh copper grids (EMS) and stained with a filtered lead-citrate solution (80 mM lead nitrate, 120 mM sodium citrate pH 12) for 2 min at RT. Sections were viewed in a CM100bio TEM (FEI, Eindhoven). Contact between GFP-μNS or Atg11-GFP-μNS particles with the vacuolar membrane was analyzed by manual quantification of randomly selected cell profiles on three independent grids per conditions.

**Statistics and reproducibility**. To assess statistical significance, two-tailed unpaired $t$-tests were performed. The underlying data of box plots and bar charts presented in this study and the exact $p$ values, means, and standard deviations are provided in the source data.

**Reporting summary**. Further information on research design is available in the Nature Research Reporting Summary linked to this article.

## Data availability

All relevant data supporting the key findings of this study are available within the article and its Supplementary Information files or from the corresponding author upon reasonable request. Raw data underlying the bar graphs and uncropped western blots are provided in the Source Data file. Source data are provided with this paper.

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

## Acknowledgements

We would like to thank Dominik Kaiser for help with insect cell expression, Irmgard Fischer and the Max Perutz Labs BioOptics facility for help with microscopy, Thomas Leonard and Linda Trübestein for help with insect cell culture, Serge Pelet for plasmids, Chris Meisinger for antibodies, Daniel Klionsky for the multiple-knock out strain and antibodies, and Life Science Editors (Angela Andersen) for help with paper editing. Electron microscopy imaging was performed at the UMCG microscopy and imaging center (UMIC). The Kraft laboratory has received funding from the Deutsche Forschungsgemeinschaft (DFG, German Research Foundation), Project ID 409673687 (to C.K.); SFB 1381 (Project-ID 403222702; to C.K.); SFB 1177 (Project-ID 259130777; to C.K., N.C.); under Germany's Excellence Strategy (CIBSS-EXC-2189- Project ID 390939984; to C.K.); from the European Research Council (ERC) under the European Union's Horizon 2020 research and innovation programme (grant agreement No 769065, to C.K., D.M.H., and M.L.); from the FWF Austrian Science Fund (grant number P25522-B20; to C.K., D.S., and W1261, to C.K., D.M.H.). This work was further supported by ENW KLEIN-1 (OCENW.KLEIN.118; to F.R.) and ZonMW TOP (91217002, to F.R.) grants, by an ALW Open Programme (ALWOP.355; to M.M.), and by a Marie Skłodowska-Curie ETN grant under the European Union's Horizon 2020 Research and Innovation Programme (Grant Agreement No 765912, to C.K., M.L., and F.R.). This work reflects only the authors' view and the European Union's Horizon 2020 research and innovation programme is not responsible for any use that may be made of the information it contains.

## Author contributions

Conceptualization D.M.H., M.L., and C.K.; Methodology D.M.H., M.L., M.M., F.R., H.M and C.K.; Validation D.M.H., M.L., N.K., D.S., M.M., F.R., H.M., and C.K.; Investigation D.M.H., M.L., N.K., D.S., H.M. and M.M.; Writing—Original Draft C.K.; Writing—Review and Editing D.M.H., M.L., F.R. and C.K.; Visualization D.M.H., M.L., M.M. and C.K.; Supervision F.R. and C.K.; Project Administration D.M.H., M.L., and C.K.; Funding Acquisition F.R. and C.K.

## Funding

## Competing interests

The authors declare no competing interests.

## Additional information

**Peer-review information** *Nature Communications* thanks Kuninori Suzuki, Thomas Wollert and the other anonymous reviewer(s) for their contribution to the peer review of this work. Peer-reviewer reports are available.

