## [Peer Review File · Nature Communications]

Reviewers' Comments:

Reviewer #1:

Remarks to the Author:

The manuscript entitled "Spatiotemporal control of avidity regulates initiation and progression of selective autophagy." by Hollenstein et al. aims to analyze the initiation of selective autophagy by focusing on the putative role of Vac8 at the vacuolar membrane in *Saccharomyces cerevisiae*.

Major points: In the following, I will comment on the major claims and experiments of the paper:

1) "Here we show that Vac8 acts as a central hub to nucleate the phagophore assembly site (PAS) at the vacuolar membrane during selective autophagy. Vac8 directly recruits the cargo complex via the Atg11 scaffold."

- Comment: It has been demonstrated before that Vac8 is involved in the formation of the PAS (e.g. Gatica et al. *Autophagy* 2020b; Fujioka et al. *Nature* 2020 and the authors themselves: Hollenstein et al. *J. Cell. Sci.* 2019). While the authors do find an interaction of Vac8 to Atg11, they do not explain the relation to the well established Vac8 – Atg13 interaction (e.g. Scott et al. *J. Biol. Chem.* 2000; Gatica et al. *Autophagy* 2020a; Fujioka et al. *Nature* 2020), which is also required for this process. Therefore, the epistasis of these two interactions is unsolved and with this the functional relevance to the molecular mechanism of the formation of the PAS is also not clear.

2) "In addition, Vac8 recruits the PI3K complex independently of autophagy. Cargo dependent clustering and Vac8-dependent sequestering of these early autophagy factors, along with local Atg1 activation, promote PAS assembly at the vacuole."

- Comment: The functional interaction of Vac8 and the PI3K complex has been demonstrated before as part of the Tor pathway (e.g. Zurita-Martinez et al. *Genetics* 2007). The experiments presented by the authors do not exclude that another factor of the Vac8/Atg13/Atg1-dependent PAS complex is directly required for the targeting of the PI3K complex. Therefore, it is unclear whether Vac8 is directly or only indirectly involved.

3) "Importantly, ectopic Vac8 redirects autophagosome formation to the nuclear membrane, indicating that the vacuolar membrane is not specifically required. We propose that multiple avidity-driven interactions drive the initiation and progression of selective autophagy."

- Comment: Again (see point 1), the central requirement of Vac8 for the assembly of the PAS and the following autophagosome formation is established. It is also known that different organelles, depending on the experimental setup, can function as membrane source for the autophagosome formation. Therefore, this result supports previous findings.

Minor points:

- Please show the standard deviation for all statistical data (Fig. 1B, Fig. 2B, Fig. 2D, Fig. 2F, Fig. 3C, Fig. 3E, Fig. S3B, Fig. 4B, Fig. 4C, Fig. 4D, Fig. 5A, Fig. 5B, Fig. 5D, Fig. S5C, Fig. 6B, Fig. 6C, Fig. 6D, Fig. 6E, Fig. 7B.).

- Please analyze the statistical significance in the differences reported in Fig. 2B, Fig. 4B, Fig. 4C, Fig. 4D, Fig. 5B, Fig. 5D, Fig. S5C, Fig. 7B.

In summary, the experiments indicate interesting clues for the research on the PAS formation in *S. cerevisiae*. However, the manuscript is not suitable for publication in *Nat. Commun.* in its current state. I hope the authors find the comments helpful in order to solve the mentioned issues.

Reviewer #2:

Remarks to the Author:

Two modes of autophagy have been known so far. One is so-called non-selective autophagy, and the other is selective autophagy. In the yeast, *Saccharomyces cerevisiae*, non-selective autophagy

starts from the formation of the bulk PAS (phagophore or pre-autophagosomal structure) by liquid-liquid phase separation upon starvation. On the other hand, the formation of the selective PAS requires cargo-receptor complex. In this manuscript entitled as "Spatiotemporal control of avidity regulates initiation and progression of selective autophagy" by Hollenstein et al., the authors find that selective PAS assembly is a consequence of sequential clustering of Atg proteins to the site where the cargo-receptor complex and scaffold complex for selective autophagy interacts. Furthermore, they find that selective autophagy can be reconstituted on the nuclear membrane other than the vacuolar membrane.

Although the data are convincing and sound, I think that the advancement achieved in this study is not sufficient for publication in this high-profile journal. The reviewer feels that a few major concerns should be solved to become publishable in Nature Communications. Without this revision, this paper would not appeal to the readership of this journal.

Major concerns

1. The bulk PAS is a liquid droplet formed by liquid-liquid phase separation. In this manuscript, there are no experiments trying to clarify whether the selective PAS is a liquid droplet or not. The reviewer strongly recommends the authors to add experiments to discuss such a possible feature of the selective PAS.
2. In this study, μ NS is used to form clusters of tagged proteins. GFP- μ NS forms globular cytoplasmic particles, which behave as genetically encoded intracellular beads (Parry et al., 2014). Although the reviewer feels it clear that Atg11-GFP- μ NS forms a cluster which associates with the vacuolar membrane, the reviewer wonders whether this cluster maintains the characteristics of endogenous Atg11. At least, the authors should show whether Atg11-GFP- μ NS keeps its activity by expressing it in *atg11 Δ* cells. Otherwise, the authors should show the oligomeric state of Atg11-GFP- μ NS is similar to that of Atg11 in some way.
3. In the last sentence of page 6, the authors conclude that "these data demonstrate that Vac8 is required to recruit Atg11 complexes to the vacuolar membrane". However, in my eyes, these data just demonstrate that Vac8 is required to recruit (artificially formed) Atg11 clusters to the vacuolar membrane. I strongly recommend the authors to address the mechanism of recruitment of Atg11 clusters without the artificial clustering system.

Specific comments

4. The reviewer feels it problematic that there are no statistical tests throughout this manuscript.
5. A part of Figures 1A and S1 has already published by other groups (Shintani et al. 2002; Suzuki et al. 2002). The authors should mention these papers.
6. In Figure 1C, only one cell for each strain is shown. The authors should quantify the association of μ NS-tagged proteins with the vacuolar membranes.
7. In Figure 1D & 1E, Atg11-GFP- μ NS associates with the vacuolar membrane. In this context, there is a possibility that this association is mediated by Atg13. The authors should examine the vacuole localization of Atg11-GFP- μ NS in the absence of Atg13.
8. In Figure 2A, similar results have been shown by other groups (Shintani et al., 2002; Fujioka et al., 2020). Please refer properly.
9. In Figure 2B, the defect in targeting of Ape1 to the vacuolar membrane depends on Atg11 or Vac8. Moreover, the defect in *atg11 Δ* cells looks severer than that in *vac8 Δ* cells. Please discuss why the levels of Ape1 targeting to the vacuole are different between *atg11 Δ* and *vac8 Δ* cells. Furthermore, the defect in *atg11 Δ vac8 Δ* cells is a similar level with that in *vac8 Δ* cells. This indicates that VAC8 is epistatic to ATG11 in vacuolar targeting of Ape1. The reviewer recommends the authors to add experiments to explain the reason why Ape1 can become targeted to the vacuolar membrane by additional deletion of ATG11 in *vac8 Δ* cells.
10. The authors clearly show that Atg11 directly interacts with Vac8 by the pull-down assay in Figure 3A. However, there is a possibility that Atg13 enhances the interaction between Atg11 and Vac8 in vivo. The reviewer recommends the authors to add experiments using ATG13-deleted cells in Figure 2G.
11. There are some typographical errors of "oligomere" (six in the text, and at least seven in Figures).

Reviewer #3:

Remarks to the Author:

This study addresses the important question how selective autophagy is initiated in yeast. The authors used an elegant approach to reconstitute the earliest step of selective autophagy in the cellular context which allowed them to dissect the mechanism of autophagy initiation. The developed reconstitution approach is based on synthetic clustering of the autophagy scaffold Atg11 by fusing it to the reoviral nonstructural protein μ NS as well as targeting of Atg11 or Vac8 to lysosomes and the nucleus. The authors found that clustering of Atg11 and Vac8 at the vacuolar membrane is required for their efficient interaction and to initiate autophagosome formation. The major conclusion of the manuscript is that avidity driven interactions of Atg11 and Vac8 are required to recruit downstream Atg proteins in order to initiate the formation of functional autophagosomes. Overall, this is a very elegant study using a creative experimental approach. However, some of the conclusions are drawn by indirect interpretations rather than direct observations as explained in more detail in the following comments. These points need to be clarified and additional experiments will be required to maintain all claims of this manuscript. This is particularly important for data claiming that avidity caused by clustering of Atg11 and Vac8, is required to induce selective autophagy. Furthermore, the manuscript lacks time-dependent analysis and the title "spatiotemporal control" should thus not be used. A revised paper would deserve publication in Nat. comm.

- 1) The authors found that Atg11-GFP- μ NS clusters at the vacuolar membrane in the absence of Atg19, arguing that the process is independent of autophagy receptors. An interesting question would be whether Atg19 is recruited to Atg11-GFP- μ NS puncta if it cannot bind its cargo Ape1 and whether this impacts on clustering of Atg11. Other selective autophagy receptors that are still expressed in Δ atg11 Δ atg19 cells might compensate for the deletion of Atg19. The authors could test whether other receptors colocalize with Atg11 on vacuolar membranes.
- 2) The expression of Atg11-GFP- μ NS induces the formation of large aggregates. The authors use the term oligomers, which might be misleading since it suggests an ordered structure with a limited, well-defined number of molecules. Atg11-GFP- μ NS aggregates are, however, 1 μ m in diameter. These aggregates are surrounded by mCherry-Vac8 and the authors concluded that the vacuolar membrane wraps around these aggregates maybe by a zipper like mechanism (Figure 1D). However, such zippering is not seen in EM (Figure 2C). Are the authors sure that the aggregates seen in Figure 1D are outside and not in the vacuolar lumen? A similar observation can be made for the Atg11 truncate 1-607 (Figure S2C), which localizes not to the surface, but to the lumen of the vacuole.
- 3) The authors found that clustering of Vac8 by expressing ot-Vac8 Δ N is not sufficient to recruit GFP-ATG11. They concluded that clustering of both proteins is required for their interaction. However, given that Atg11-GFP- μ NS forms large aggregates rather than oligomers, the clustering of Atg11-GFP- μ NS at the vacuolar membrane might be an effect caused by aggregation and not a specific step during the initiation of autophagy. The authors concluded that the process is avidity driven, but this conclusion is not fully supported until an indirect effect of Atg11 aggregation can be excluded. This is particularly true because control constructs (GFP- μ NS) form much smaller puncta.
- 4) If Vac8 is clustered by its recruitment to Ape1 complexes and Atg11 by its fusion to μ NS, both puncta colocalize and the authors presented this as a proof of avidity (Figure 3 F and G). In this scenario, one would expect to see two puncta 'touching' each other. However, the authors observed that GFP fluorescence was entirely surrounded and embedded by BFP. An alternative interpretation of the data would be that Vac8 recruits Atg11 independently of clustering and that clustering of Atg11 follows initial recruitment. Although the authors did a control experiment using GFP-Atg11 (not tagged with μ NS), the two experiments are different. In Figure 3D Atg11 is tagged at its N-term whereas in Figure 3F it is C-terminally tagged. The authors need to repeat this critical experiment with only one type of fusion protein to exclude that tagging impairs protein function. Furthermore, it might be difficult to reveal GFP-Atg11 fluorescence in the absence of clustering on ot-Vac8 Δ N dots due to lower abundance and weaker signal. The authors should use biochemical assays to prove that clustering of both proteins is required for efficient interaction. This is particularly important since an interaction of both proteins was observed using co-IP of PA-Atg11 with GFP-Vac8 (Figure 2G).
- 5) The assembly of the PAS involves an ordered recruitment of the Atg protein machinery. To test, whether Vac8 is important for initiation, elongation/maturation or fusion, the authors tested at

which stage the recruitment-cascade was disrupted if Vac8 was deleted. They found that puncta formation of Atg8 and Atg14 was reduced in *vac8Δ* cells, while Atg9 was still recruited to Ape1. The authors further found that the PI3K complex was recruited by Vac8 to the vacuolar membrane independently of its function in autophagy. It thus remains unclear whether the reduction in Atg14 puncta formation observed in *vac8Δ* cells is related to autophagy or to the autophagy independent interaction of both. If this alternative interpretation is true, puncta formation of Atg14 is only a consequence of Vac8 clustering, which is consistent with data in Figure 5A, B. It remains thus unclear at which stage autophagy is blocked in *vac8Δ* cells and the conclusions that it is involved in maturation is not supported by data.

6) In Figure 5C, it is shown that targeting of Atg14 to the vacuole is sufficient to recruit Atg11/Ape1 complexes in the absence of Vac8. Why do the authors conclude that both, tethering of Atg11 and Atg14 at the vacuole is required for PAS formation, if tethering of Atg14 seems to be sufficient as indicated by the data. Altogether, it appears that clustering of Vac8 at the vacuolar membrane recruits Atg14 which in turn recruits Atg11. The argument that Atg11 induces clustering of Vac8 (Figure 1D and S2D) is not very convincing.

7) The authors reconstituted the assembly of the PAS ectopically in vivo by targeting Vac8 to the nucleus. In Figure 6B, Vac8 clusters independently of Atg11 cargo complexes on nuclear membranes, while at Vac8-Atg11 contact sites no clustering of Vac8 was observed. This suggests that the approach does not fully recapitulate PAS assembly. Furthermore, the colocalization of Vps15 with nucleus targeted Vac8 is not convincing. The authors need to analyze protein recruitment by analyzing protein levels in nuclear membrane fractions by western-blotting. Interestingly, Atg2 puncta colocalize with Vac8 clusters at the nucleus, suggesting that PAS assembly does not depend on Atg11 clustering. The authors should analyze recruitment of Atg2 by nt-Vac8 in *atg11Δ* cells.

8) The recruitment of Atg2 to the "nuclear PAS" appears to depend on Atg14 and the authors concluded that PI3P production at the vacuole is required for this process. However, an alternative interpretation could be that clustering of Vac8 leads to an increase in local Atg2 which is easier to detect (Figure S7C). If the authors want to maintain the conclusion that PI3P is produced at the nuclear membrane, a direct assay would be required (e.g. PI3P sensor colocalization).

9) The observation that Ape1 is transported to the vacuole in *vac8Δ* cells expressing nt-Vac8 (Figure 7A) could also rely on tight nuclear-vacuolar contact sites as present in Figure 7A. The conclusion that PAS initiation at the nucleus leads to the formation of functional autophagosomes that deliver their content to the vacuole is thus not supported by this Figure. This conclusion would be significant and important, but more convincing data are needed. E.g. blocking fusion of autophagosomes with the vacuole will lead to an accumulation of Atg8-positive Ape1 puncta in *vac8Δ* cells expressing nt-Vac8 but not in cells lacking Vac8.

Response to the points raised by the referees

We thank all three referees for their insightful comments, which helped us to substantially improve our manuscript. As explained in detail below, we have addressed all points and added extensive new validation data, which further support our conclusions.

Reviewer #1 (Remarks to the Author):

The manuscript entitled “Spatiotemporal control of avidity regulates initiation and progression of selective autophagy.” by Hollenstein et al. aims to analyze the initiation of selective autophagy by focusing on the putative role of Vac8 at the vacuolar membrane in *Saccharomyces cerevisiae*.

Major points: In the following, I will comment on the major claims and experiments of the paper:

1) “Here we show that Vac8 acts as a central hub to nucleate the phagophore assembly site (PAS) at the vacuolar membrane during selective autophagy. Vac8 directly recruits the cargo complex via the Atg11 scaffold.”

- Comment: It has been demonstrated before that Vac8 is involved in the formation of the PAS (e.g. Gatica et al. Autophagy 2020b; Fujioka et al. Nature 2020 and the authors themselves: Hollenstein et al. J.Cell. Sci. 2019). While the authors do find an interaction of Vac8 to Atg11, they do not explain the relation to the well established Vac8 – Atg13 interaction (e.g. Scott et al. J. Biol. Chem. 2000; Gatica et al. Autophagy 2020a; Fujioka et al. Nature 2020), which is also required for this process. Therefore, the epistasis of these two interactions is unsolved and with this the functional relevance to the molecular mechanism of the formation of the PAS is also not clear.

*We agree with the reviewer that the relation of Vac8 and Atg13 is an important aspect and realize we did not explain this carefully enough in the manuscript. In this manuscript, we looked at the role of Vac8 in **selective** autophagy, which is different from its role in **bulk** autophagy. The mentioned studies all address Vac8 in **bulk** autophagy, however, its role in **selective** autophagy has not been analyzed yet. Also, the function of Atg13 differs significantly between selective and bulk autophagy. In selective autophagy, PAS formation is triggered by the assembly of the core autophagy machinery on cargo-receptor complexes. Atg13 brings Atg1 to the vacuole to enable Atg1's recruitment to these cargo-receptor complexes (Torggler et al. Mol Cell 2016).*

*Bulk autophagy does not require cargo to initiate PAS formation. Here Atg13 plays an essential role in generating a scaffold/phase separated structure, the bulk PAS, which then serves as an assembly platform for the other autophagy factors. Furthermore, we have previously shown that the only role of Atg13 in selective autophagy is Atg1's recruitment to the vacuole (Torggler et al., Mol Cell 2016). **As Atg13 can be bypassed in selective autophagy (by synthetic tethering of Atg1 to the vacuole), the interaction between Atg13 and Vac8 is only required for bringing Atg13 and therefore Atg1 to the vacuole, but Atg13 does not serve additional essential functions.** This is different from the absolutely essential role of Atg13 in bulk autophagy.*

To further validate that Atg13 is not required in recruiting Atg11-cargo to the vacuole, we have addressed the relationship of Atg13, Atg11 and Vac8 by showing that in the absence of Atg13 the Atg11-cargo complex is still recruited to the vacuole (Fig. 1b).

In addition, we showed that oligomer tethered Vac8 and Atg11- μ NS can interact in the absence of Atg13 (Fig. 3g).

We have substantiated these findings further: In the updated Fig. 1d and 1e, we now show that Atg13 is dispensable for Atg11- μ NS vacuole deformation. In the new Supplementary Fig. 3c, we furthermore show that the Vac8-Atg11 interaction is not promoted by Atg13.

Figure 1d and 1e

Supplementary Figure 3c

Together, these findings further underline how the role of Atg13 in selective and bulk autophagy is different, and that also the interaction of Vac8 with Atg13 serves a different function in these two types of autophagy.

Furthermore, how Vac8 acts in **selective** autophagy, why the vacuolar proximity of the PAS is required, and how the PAS factors assemble in a regulated manner has remained completely elusive. Here we clarify these points in **selective** autophagy and show that the function of Vac8 is essential, and different from its Atg13-dependent role in bulk autophagy. Therefore, we strongly believe our study is highly novel and significantly contributes to the understanding of molecular mechanisms in **selective** autophagy.

We explain this difference in the introduction, results and the discussion parts:

Introduction: “Atg13, however, is dispensable for the recruitment of Atg11-bound cargo complexes to the vacuole (Torggler et al., 2016) and thus dispensable for the initiation of selective PAS formation. It therefore remains unclear which role Vac8 plays during selective autophagy.”

Results: “To determine if other Atg proteins are required to anchor Atg11 at the vacuole, we tested seven Atg proteins from the main functional groups: Atg1 and Atg13 ... In contrast, BFP-Ape1 was recruited to the vacuole in atg1 Δ , atg13 Δ , atg9 Δ , atg14 Δ , atg12 Δ , atg2 Δ and atg8 Δ mutants, similar to wild type cells (Fig. 1a, 1b and Supplementary Fig. 1a). Thus, canonical Atg proteins are not required to anchor Atg11-bound cargo complexes at the vacuole.”

These experiments also show avidity dependent interaction between Vac8 and Atg11 when Atg13 is deleted.

Discussion: we explain the function of the Vac8-Atg13 interaction: “Vac8 interacts with Atg13, which recruits and confines the Atg1 kinase to the vacuolar membrane. This allows Atg1 to bind Atg11-bound to cargo complexes, resulting in clustering-induced trans-autophosphorylation and local kinase activation at the vacuole (Kamber et al., 2015; Torggler et al., 2016).”

2) “In addition, Vac8 recruits the PI3K complex independently of autophagy. Cargo dependent clustering and Vac8-dependent sequestering of these early autophagy factors, along with local Atg1 activation, promote PAS assembly at the vacuole.”

- Comment: The functional interaction of Vac8 and the PI3K complex has been demonstrated before as part of the Tor pathway (e.g. Zurita-Martinez et al. Genetics 2007). The experiments presented by the authors do not exclude that another factor of the Vac8/Atg13/Atg1-dependent PAS complex is directly required for the targeting of the PI3K complex. Therefore, it is unclear whether Vac8 is directly or only indirectly involved.

The mentioned study found a synthetic growth defect between tor1 mutants and vps15 Δ and vps34 Δ deletions. However, they also found that a tor1 Δ atg13 Δ double-mutant strain did not show a defect, suggesting that the synthetic lethality between vps15 Δ /34 Δ and tor1 Δ mutants is not due to a defect in autophagy. There is no further investigation of the relationship of Vac8 and the PI3K in that study.

Although it has been shown earlier in other studies that both Vac8 and PI3K complex members are required for autophagy function, to our knowledge no physical and no functional interaction between the PI3K and Vac8 have been reported prior to the submission of our manuscript. Also, it was completely unclear how the PI3K complex is recruited to the vacuole.

However, it is important to distinguish between 1. vacuole targeting of the PI3K and 2. its association with the PAS, which are two different subsequent events happening, both required for selective autophagy function.

1. Vacuolar recruitment of the PI3K complex by Vac8 is direct and does not require Atg11 (Fig. 4e, 4f and Supplementary Fig. 6b). To strengthen this notion, we used the

multiple knock-out (MKO) strain created by the Klionsky lab, which lacks all core Atg proteins (including Atg11, Atg1 and Atg13), but Vac8 is present. In this strain, expression of Atg14-GFP together with its partner Vps30 (also called Atg6) resulted in vacuolar localization, further supporting that core autophagy factors are dispensable for the Vac8-dependent recruitment of the PI3K to the vacuole (new Supplementary Fig. 6d). (Note that Atg14 needs to be co-expressed with Vps30, as the absence of Vps30 also destabilizes Atg14, Kihara et al., JCB 2001)

Supplementary Figure 6d

2. The Reviewer raises an important point in pointing out that Vac8 is not directly involved in targeting the PI3K to the PAS. We did not intend to claim this and apologize for not making this clear. What we show is that Vac8 is **indirectly** involved, by recruiting the PI3K to the vacuole, which is a prerequisite for targeting the PI3K to the PAS (via avidity).

It has previously been shown that in selective autophagy, formation of the early PAS complex and its vacuolar localization do not depend on Atg13 and Atg1 (Torggler et al., 2016, Mol. Cell FigS3E and S6D). We and others have also shown that Atg14 depends on Atg9 for its recruitment to the PAS (Fig. 4c, and He et al., MBoC 2008). In contrast, in the absence of Atg11 or Atg19 no selective autophagy PAS forms and hence no selective autophagy takes place.

To further substantiate that no factor of the Vac8/Atg13/Atg1-dependent PAS complex is directly required for the targeting of the PI3K complex, we analyzed Atg14-GFP PAS association also in cells lacking Atg1 and Atg13 (new Supplementary Fig. 5c and 5d). As expected from the literature, in the absence of Atg1 and Atg13, Atg14-GFP forms PAS puncta in fluorescence microscopy, supporting that these factors are not required. However, in the absence of the cargo receptor Atg19, which results in a failure of cargo targeting to the vacuole, no selective PAS and therefore no Atg14-GFP puncta are observed, further supporting that vacuolar localization of the cargo complex is required for the PI3K PAS association, rather than components of the Atg1 kinase complex.

We have made adjustments to the text to better explain the literature and included the new figures.

Supplementary Figure 5c and 5d

3) “Importantly, ectopic Vac8 redirects autophagosome formation to the nuclear membrane, indicating that the vacuolar membrane is not specifically required. We propose that multiple avidity-driven interactions drive the initiation and progression of selective autophagy.”

– Comment: Again (see point 1), the central requirement of Vac8 for the assembly of the PAS and the following autophagosome formation is established. It is also known that different organelles, depending on the experimental setup, can function as membrane source for the autophagosome formation. Therefore, this result supports previous findings.

*As mentioned above and as stated by the reviewer, the role of Vac8 in establishing the bulk PAS via Atg13 has been well established. However, how Vac8 functions in **selective** autophagy has been completely unknown.*

*In this study we address the role of Vac8 in selective autophagy initiation, and the role of the vacuole in this process. We agree that many membranes have been implicated as a lipid source for providing lipids for autophagosome formation, however, this is not the question we have addressed in this work. PAS formation precedes autophagosome formation (and therefore lipid source requirement), and has been observed to happen at the vacuole. It has remained elusive why PAS formation happens at the vacuole and if this localization is actually needed. Also, it has been totally unclear how the connection between the Atg11-cargo and the vacuole is established. **Here we show (i) that the direct Vac8-Atg11 interaction establishes this connection, (ii) that this connection is essential for selective autophagy function, and (iii) that tethering of autophagy factors to the vacuolar membrane is required to increase their effective local concentration to then allow an avidity-driven assembly of the PAS.***

*As selective autophagy can be reconstituted also on the nuclear membrane, we rather propose that **the vacuole per se is not needed**. In fact, another/any membrane assembly platform can serve this task, if Vac8 is relocated there to coordinate the assembly. We do not address the role of the vacuolar membrane as a membrane source, but our results rather propose that it does not contribute as such.*

Altogether, our findings are novel and clarify two questions in the field, which had remained unclear for a long time: 1. That the selective PAS - vacuole connection is established by the direct Vac8-Atg11 interaction and 2. that selective PAS assembly is regulated by avidity. We have added further support for these findings in the revised version of the manuscript.

Minor

points:

- Please show the standard deviation for all statistical data (Fig. 1B, Fig. 2B, Fig. 2D, Fig. 2F, Fig. 3C, Fig. 3E, Fig. S3B, Fig. 4B, Fig. 4C, Fig. 4D, Fig. 5A, Fig. 5B, Fig. 5D, Fig. S5C, Fig. 6B, Fig. 6C, Fig. 6D, Fig. 6E, Fig. 7B.).

*As requested by the reviewer, we provide descriptive statistical data in **"Quantification_and_statistical_tests.xlsx"** in the source data.*

- Please analyze the statistical significance in the differences reported in Fig. 2B, Fig. 4B, Fig. 4C, Fig. 4D, Fig. 5B, Fig. 5D, Fig. S5C, Fig. 7B.

*As requested by the reviewer, we provide analysis of statistical significance in **"Quantification_and_statistical_tests.xlsx"** in the source data.*

In summary, the experiments indicate interesting clues for the research on the PAS formation in *S. cerevisiae*. However, the manuscript is not suitable for publication in *Nat. Commun.* in its current state. I hope the authors find the comments helpful in order to solve the mentioned issues.

We believe we have addressed and clarified all points raised by the reviewer.

Reviewer #2 (Remarks to the Author):

Two modes of autophagy have been known so far. One is so-called non-selective autophagy, and the other is selective autophagy. In the yeast, *Saccharomyces cerevisiae*, non-selective autophagy starts from the formation of the bulk PAS (phagophore or pre-autophagosomal structure) by liquid-liquid phase separation upon starvation. On the other hand, the formation of the selective PAS requires cargo-receptor complex. In this manuscript entitled as “Spatiotemporal control of avidity regulates initiation and progression of selective autophagy” by Hollenstein et al., the authors find that selective PAS assembly is a consequence of sequential clustering of Atg proteins to the site where the cargo-receptor complex and scaffold complex for selective autophagy interacts. Furthermore, they find that selective autophagy can be reconstituted on the nuclear membrane other than the vacuolar membrane. Although the data are convincing and sound, I think that the advancement achieved in this study is not sufficient for publication in this high-profile journal. The reviewer feels that a few major concerns should be solved to become publishable in *Nature Communications*. Without this revision, this paper would not appeal to the readership of this journal.

Major concerns

1. The bulk PAS is a liquid droplet formed by liquid-liquid phase separation. In this manuscript, there are no experiments trying to clarify whether the selective PAS is a liquid droplet or not. The reviewer strongly recommends the authors to add experiments to discuss such a possible feature of the selective PAS.

This is an interesting point raised by the reviewer. Phase separation has been reported for the formation of the supramolecular structure of Atg1/13/17-31-29, which is the earliest event of initiating the PAS in bulk autophagy. For this phase separation the disordered region of Atg13 as well as the Atg17-31-29 complex have been reported to be absolutely essential.

PAS assembly in selective autophagy does not require such a supramolecular structure because the cargo complex serves as an assembly platform, Atg13 can be completely bypassed in selective autophagy (Torggler et al., Mol Cell 2016) and Atg11 has no predicted disordered regions that would promote phase separation. Also, the Atg17-31-29 complex is dispensable for the selective Cvt pathway. Thus, we believe it is highly unlikely that phase separation of Atg proteins drives early PAS assembly also in selective autophagy.

It has indeed been shown that the selective cargo Ape1 does phase separate: A commonly performed experiment to investigate phase separation is the application of a compound that disperses phase separated structures, such as 1,6-hexanediol. Upon 1,6-hexanediol treatment the Ape1 cargo puncta were dissolved in vivo (Yamasaki et al. 2020). As treatment with 1,6-hexanediol disperses the phase separated cargo, thereby preventing the assembly of the PAS (which requires a cargo to form), any further analysis in vivo is impossible. To address the question if the selective autophagy PAS by itself also undergoes phase separation, one would require reconstituting the cargo and the complete PAS in vitro, which is beyond the scope of this manuscript. We appreciate that this is an interesting question and mention this aspect in the discussion.

2. In this study, μ NS is used to form clusters of tagged proteins. GFP- μ NS forms globular cytoplasmic particles, which behave as genetically encoded intracellular beads (Parry et al., 2014). Although the reviewer feels it clear that Atg11-GFP- μ NS forms a cluster which associates with the vacuolar membrane, the reviewer wonders whether this cluster maintains the characteristics of endogenous Atg11. At least, the authors should show whether Atg11-GFP- μ NS keeps its activity by expressing it in *atg11 Δ* cells. Otherwise, the authors should show the oligomeric state of Atg11-GFP- μ NS is similar to that of Atg11 in some way.

Atg11 fulfills multiple essential roles during initiation of selective autophagy by interacting with several different proteins: Binding to cargo receptors in selective autophagy such as Atg19, bringing the cargo to the vacuole and recruitment of autophagy factors such as Atg1 and Atg9. With the use of Atg11- μ NS we wanted to exclusively address the vacuole recruitment function, independent of cargo and cargo-receptors, and we did not aim to generate a fully functional Atg11 allele. We showed that Atg11's vacuolar recruitment function is fully operational, also when clustered on μ NS (Fig. 1d).

To further verify that Atg11 fused to μ NS retains its functional properties, we have now performed additional experiments: In the new Supplementary Fig. 1c, 1d, we show that Atg11- μ NS particles are capable of recruiting the cargo receptor Atg19, Atg1 and Atg9. In addition, Atg11- μ NS is still capable of homo-oligomerizing, as cytosolic Atg11 is recruited to these particles. Therefore, artificial tethering of Atg11 to μ NS particles does not impair its natural interactions.

Supplementary Figure 1c

*However, we would not expect these Atg11- μ NS particles to complement an *atg11 Δ* in CVT pathway function, as the high number of Atg11- μ NS molecules will likely titrate away Atg proteins required for Ape1 targeting in the Cvt. We nevertheless checked CVT pathway progression in wild type and *atg11 Δ* cells containing Atg11- μ NS. We observed that Atg11- μ NS has a dominant negative effect also in wild type cells, suggesting that indeed Atg11- μ NS particles titrate away CVT pathway factors.*

However, some Ape1 processing could be observed (Reviewer Figure 1), similarly in wild type and *atg11Δ* cells, suggesting that Atg11-μNS can still promote CVT pathway function to some degree. Together, these observations support that Atg11-μNS retains its vacuole targeting and binding properties and is suitable for our studies.

It should be noted that the structure of μNS particles has not been solved yet (Broering et al., J Virology 2002, McCutcheon et al., Virology 1999). Therefore, it remains unclear if μNS forms oligomers or aggregates. We therefore changed the terminology to "μNS particles" throughout the manuscript.

Reviewer Figure R1: Functionality of Atg11-μNS particles in the CVT pathway, in wild type, *atg11Δ* and *atg1Δ* cells, monitored by Ape1 processing.

3. In the last sentence of page 6, the authors conclude that “these data demonstrate that Vac8 is required to recruit Atg11 complexes to the vacuolar membrane”. However, in my eyes, these data just demonstrate that Vac8 is required to recruit (artificially formed) Atg11 clusters to the vacuolar membrane. I strongly recommend the authors to address the mechanism of **recruitment of Atg11 clusters without the artificial clustering system**.

We agree with the reviewer that this is an important point to address. We indeed analyzed native Atg11-cargo complex recruitment in wild type and *vac8Δ* cells and found that *vac8Δ* cells are defective in vacuolar recruitment of Atg11-cargo complexes (Fig. 2a and b). As a control we used *atg11Δ* cells, as Atg11 is known to be absolutely required for vacuole targeting of the Ape1 cargo (Suzuki Dev Cell 2002, Shintani Dev Cell 2002). There is an apparent increase in vacuole binding of cargo in *vac8Δ* cells. This stems from the fragmentation phenotype of *vac8Δ* mutants, resulting in the vacuole taking up a bigger space of the cytosol, which makes the analysis of cargo localization to the vacuole more difficult (see Reviewer Figure R2 below for examples of this fragmentation). However, we saw no additional defect by deleting ATG11 on top of VAC8. This highly suggests that Vac8 together with Atg11 mediates the vacuole recruitment of the cargo, and that the increased vacuolar localization in *vac8Δ* and *atg11Δ vac8Δ* mutants does not stem from real vacuolar binding, but rather from the difficult analysis in cells displaying a high fragmentation of the vacuole.

We appreciate that this was not fully clear and clarified it in the revised text.

Reviewer Figure R2: Examples of vacuolar fragmentation in *vac8Δ* cells.

To strengthen this finding, we verified the role of Vac8 and Atg11 in vacuolar attachment by EM and observed a complete loss of Atg11-uNS vacuole attachment in vac8Δ cells (Fig. 2c).

There are two problems in analyzing endogenous Ape1 oligomers by EM. This technique examines thin cell sections and not entire cells. As a result, the frequency of finding an Ape1 oligomer in these thin sections is very low: for instance, in an atg11Δ only 0.3% of cell profiles show an Ape1 oligomer (see Mari et al., JCB 2010). For this reason, we performed the EM analysis with Atg11-μNS. As we have well characterized the usefulness of Atg11-μNS (see point 2. above), and we have verified the importance of Vac8 also with the endogenous Atg11-cargo complex by fluorescence microscopy (Fig. 2a), and we have further validated the Vac8-Atg11 interaction by biochemical approaches (Fig. 2g, 3a and 3b, Supplementary Fig. 3d and 4a), we believe that the EM results confirm our proposed model.

Specific comments

4. The reviewer feels it problematic that there are no statistical tests throughout this manuscript.

As requested by the reviewer, we provide analysis of statistical significance in "Quantification_and_statistical_tests.xlsx" in the source data.

5. A part of Figures 1A and S1 has already published by other groups (Shintani et al. 2002; Suzuki et al. 2002). The authors should mention these papers.

As requested by the reviewer, we added these references also in the results section.

6. In Figure 1C, only one cell for each strain is shown. The authors should **quantify the association** of μ NS-tagged proteins with the vacuolar membranes.

As suggested by the reviewer, we quantified the association of μ NS tagged proteins with the vacuole. This data is shown in new Fig. 1e and confirms our findings.

7. In Figure 1D & 1E, Atg11-GFP- μ NS associates with the vacuolar membrane. In this context, there is a possibility that this association is mediated by Atg13. The authors should examine the **vacuole localization of Atg11-GFP- μ NS in the absence of Atg13.**

As suggested by the reviewer, we have repeated these experiments in *atg13 Δ* cells. In the absence of Atg13, Atg11- μ NS particles attach at and deform the vacuole as observed for Atg13 containing cells, supporting that Atg13 is not involved in the recruitment of Atg11- μ NS to the vacuole. We replaced the previous Figure 1c with a new figure, which also includes the analysis in *atg13 Δ* cells (new Fig. 1d and 1e).

Figure 1d and 1e

8. In Figure 2A, similar results have been shown by other groups (Shintani et al., 2002; Fujioka et al., 2020). Please refer properly.

In Fig. 2a, the *atg11 Δ* strain only serves as a negative control, however, the phenotype of the *vac8 Δ* mutant had never been analyzed before. We have cited Shintani et al. 2002 in the introduction for the requirement of Atg11:

“The cargo receptor autophagy-related 19 (Atg19) binds to both the Ape1 cargo and the scaffold protein Atg11 (Shintani et al., 2002). Subsequently, the Atg11-bound cargo is recruited to the vacuole, ...”

We added this citation now also at two places in the results section, together with Suzuki et al. 2002:

“Cargo receptors bind to Atg11, which serves as the scaffold to initiate selective autophagy. Atg11 anchors the cargo-receptor complex to the vacuole and

recruits other autophagy factors, such as Atg1, to form the PAS (ref Suzuki 2002 and Shintani 2002)."

and

"As expected, deletion of Atg11 abrogated BFP-Ape1 recruitment to the vacuole, detected by staining with FM4-64 (ref Suzuki 2002 and Shintani 2002)."

Fujoka et al. 2020 did not analyze the PAS under selective autophagy conditions, therefore we feel that this citation is not appropriate here.

9. In Figure 2B, the defect in targeting of Ape1 to the vacuolar membrane depends on Atg11 or Vac8. Moreover, the defect in *atg11Δ* cells looks severer than that in *vac8Δ* cells. Please discuss why the levels of Ape1 targeting to the vacuole are different between *atg11Δ* and *vac8Δ* cells. Furthermore, the defect in *atg11Δvac8Δ* cells is a similar level with that in *vac8Δ* cells. This indicates that VAC8 is epistatic to ATG11 in vacuolar targeting of Ape1. The reviewer recommends the authors to add experiments to explain the reason why Ape1 can become targeted to the vacuolar membrane by additional deletion of ATG11 in *vac8Δ* cells.

*This apparent difference comes from the strong fragmentation phenotype of *vac8Δ* and *atg11Δ vac8Δ* cells, which makes the accurate judgment of vacuolar proximity very difficult, resulting in a higher amount of false-positive vacuolar localization (see point 3 above). For this reason, we also analyzed the importance of Vac8 for the vacuole recruitment of Atg11-μNS by EM, which clarified that Vac8 is absolutely needed for the Atg11-cargo attachment to the vacuole. Please refer to point 3 above for a more detailed explanation.*

10. The authors clearly show that Atg11 directly interacts with Vac8 by the pull-down assay in Figure 3A. However, there is a possibility that Atg13 enhances the interaction between Atg11 and Vac8 in vivo. The reviewer recommends the authors to add experiments using ATG13-deleted cells in Figure 2G.

As proposed by the reviewer, we have performed an additional experiment to further support the finding that the Vac8-Atg11 interaction is independent of Atg13. Indeed, also in the absence of Atg13, Vac8 and Atg11 co-precipitate. These findings further support that Atg13 does not facilitate Atg11-Vac8 binding. We have included this data as new Supplementary Fig. 3c.

Supplementary Figure 3c

11. There are some typographical errors of “oligomere” (six in the text, and at least seven in Figures).

We thank the reviewer for pointing out this mistake and have corrected the typos.

Reviewer #3 (Remarks to the Author):

This study addresses the important question how selective autophagy is initiated in yeast. The authors used an elegant approach to reconstitute the earliest step of selective autophagy in the cellular context which allowed them to dissect the mechanism of autophagy initiation. The developed reconstitution approach is based on synthetic clustering of the autophagy scaffold Atg11 by fusing it to the reoviral nonstructural protein μ NS as well as targeting of Atg11 or Vac8 to lysosomes and the nucleus. The authors found that clustering of Atg11 and Vac8 at the vacuolar membrane is required for their efficient interaction and to initiate autophagosome formation. The major conclusion of the manuscript is that avidity driven interactions of Atg11 and Vac8 are required to recruit downstream Atg proteins in order to initiate the formation of functional autophagosomes. Overall, this is a very elegant study using a creative experimental approach. However, some of the conclusions are drawn by indirect interpretations rather than direct observations as explained in more detail in the following comments. These points need to be clarified and additional experiments will be required to maintain all claims of this manuscript. This is particularly important for data claiming that avidity caused by clustering of Atg11 and Vac8, is required to induce selective autophagy. Furthermore, the manuscript lacks time-dependent analysis and the title "spatiotemporal control" should thus not be used. A revised paper would deserve publication in Nat. comm.

We agree with the suggestion and have changed the title to:

"Spatial control of avidity regulates initiation and progression of selective autophagy"

1) The authors found that Atg11-GFP- μ NS clusters at the vacuolar membrane in the absence of Atg19, arguing that the process is independent of autophagy receptors. An interesting question would be **whether Atg19 is recruited to Atg11-GFP- μ NS puncta if it cannot bind its cargo Ape1 and whether this impacts on clustering of Atg11.** Other selective autophagy receptors that are still expressed in Δ atg11 Δ atg19 cells might compensate for the deletion of Atg19. The authors could test whether other receptors colocalize with Atg11 on vacuolar membranes.

We have previously observed that tethering Atg11 to Ape1 oligomers also in the absence of Atg19 is sufficient to target Ape1 to the vacuole (Torggler et al., Mol Cell 2016), therefore Atg11 is capable of vacuole binding also in the absence of cargo receptors. We want to point out that Atg11- μ NS does not cluster at the vacuolar membrane, rather the μ NS particle clusters the attached Atg11, similar to the situation found with endogenous cargo. We have shown that also in this artificial setup Atg11 is sufficient to recruit μ NS particles to the vacuole.

To further strengthen the suitability of Atg11- μ NS in our study, we now showed that Atg11 retains its natural properties, such as binding the cargo receptor Atg19, but also other autophagy proteins such as Atg1 and Atg9 and its ability for homo-oligomerization (Supplementary Fig. 1c).

Supplementary Figure 1c

As suggested by the reviewer, we have also tested if Atg11- μ NS particles are proficient in binding Atg19, in the presence and the absence of Ape1 cargo. Indeed, in both cases Atg19 associates with Atg11- μ NS particles and the particles form in a similar manner, supporting that Atg11 fused to μ NS retains its natural ability to bind to cargo receptors (new Supplementary Fig. 1d). Furthermore, we showed that the Atg11 mutant (1-454), fused to μ NS, was still proficient in binding to the vacuole. Importantly, Atg11(1-454)- μ NS lacks the C-terminal receptor binding region of Atg11 and is thus unable to recruit Atg19 or other cargo receptor proteins (Reviewer Figure 3 below, Yorimitsu et al., MBoC 2005, Aoki et al., 2011, MBoC). Hence the ability or the inability of Atg11 to interact with cargo receptors appears to have no influence on the recruitment to the vacuole.

Supplementary Figure 1d

Reviewer Figure R3: Atg19 bound to Ape1 oligomers or cytosolic Atg19 binds to Atg11- μ NS but not to Atg11(1-454)- μ NS particles.

It should be noted that μ NS particle size is heterogenous and can vary quite substantially, independently from its fusion to Atg11 (Munder et al., 2016, eLife).

Supplementary Figure 1b

2) The expression of Atg11-GFP- μ NS induces the formation of large aggregates. The authors use the term oligomers, which might be misleading since it suggests an ordered structure with a limited, well-defined number of molecules.

This is a great point raised. Actually, so far it has not been analyzed what kind of structure μ NS forms, if these are ordered oligomers or aggregates, but it seems clear that the number of molecules in this structure can vary (Broering et al., J Virology 2002, McCutcheon et al., Virology 1999). We therefore changed the wording to 'particle' throughout the manuscript.

Atg11-GFP- μ NS aggregates are, however, 1 μ m in diameter. These aggregates are surrounded by mCherry-Vac8 and the authors concluded that the vacuolar membrane

wraps around these aggregates maybe by a zipper like mechanism (Figure 1D). However, such zippering is not seen in EM (Figure 2C). Are the authors sure that the aggregates seen in Figure 1D are outside and not in the vacuolar lumen? A similar observation can be made for the Atg11 truncate 1-607 (Figure S2C), which localizes not to the surface, but to the lumen of the vacuole.

When analyzing fluorescence microscopy z-stacks, it becomes clear that these particles are only invaginated, an opening or vacuolar membrane connection towards the cytoplasm always exists. Also, in contrast to autophagic bodies, which are released into the vacuolar lumen and are rapidly moving within the vacuolar boundaries, invaginated Atg11- μ NS particles are not mobile, further supporting that a membrane connection remains.

We have tested this further, by incubating the cells in H₂O before imaging, which induces osmotic pressure and results in vacuole swelling. When incubated in H₂O, Atg11- μ NS particles that appear to be inside the vacuole are 'pushed-out' to the vacuolar surface and only attach to the outer vacuolar membrane, without invagination (Reviewer Figure R4 below). For EM analysis, cells are also incubated in H₂O, therefore, vacuolar swelling has the same effect and results in the particles appearing attached to the vacuole, we never observed invaginated particles by EM. We have now clarified this point in the revised version of the manuscript:

"Moreover, we observed a dramatic bending of the vacuolar membrane around Atg11-GFP- μ NS, suggesting that multiple interactions between Atg11-GFP- μ NS and a vacuolar binding partner deform the vacuolar membrane. Atg11-GFP- μ NS particles were not taken up into the vacuolar lumen but remained attached to the outer leaflet of the vacuolar membrane."

Reviewer Figure R4: Atg11-GFP- μ NS particles are not taken up into the vacuolar lumen. Treatment of yeast cells with H₂O results in an increase of the vacuolar turgor pressure, which counteracts vacuolar membrane deformation caused by Atg11-GFP- μ NS particles.

3) The authors found that clustering of Vac8 by expressing *ot-Vac8 Δ N* is not sufficient to recruit GFP-ATG11. They concluded that clustering of both proteins is required for their interaction. However, given that Atg11-GFP- μ NS forms large aggregates rather than oligomers, the clustering of Atg11-GFP- μ NS at the vacuolar membrane might be an effect caused by aggregation and not a specific step during the initiation of autophagy.

We would like to clarify that Atg11-GFP- μ NS does not cluster at the vacuolar membrane, rather Atg11 is clustered on the μ NS particle, reflecting the natural situation when Atg11 clusters on endogenous receptor-cargo complexes. Also in the natural situation, multiple Atg19 receptor molecules bind to the oligomeric Ape1 cargo, allowing numerous Atg11 molecules to cluster on the cargo. This clustering or local enrichment then mediates the stable interaction between Atg11 and Vac8 at the

vacuolar membrane. Also, the attachment of Atg11- μ NS at the vacuolar membrane depends on Vac8, and thus is specific for Vac8.

The authors concluded that the **process is avidity driven**, but this conclusion is not fully supported until an indirect effect of Atg11 aggregation can be excluded. This is particularly true because control constructs (GFP- μ NS) form much smaller puncta.

As suggested by the reviewer, we have performed further experiments without the use of μ NS particles to support that avidity drives the Atg11-Vac8 interaction. If the stable interaction between oligomer-clustered Atg11 and vacuolar Vac8 requires avidity, then the individual affinity between single Atg11 and Vac8 molecules must be weak. To this end, we compared the binding efficiency of Atg11 to Vac8 with that of Atg11 to Atg19 in vitro (we used Atg19-3D, which mimics phosphorylation that is required for Atg19-Atg11 interaction, see Pfaffenwimmer et al., EMBO R 2014). Indeed, the amount of Atg11 expressed in Sf9 insect cells that co-precipitated with recombinant GST-Vac8 was more than 10 times lower than with GST-Atg19, supporting a low affinity interaction between Vac8 and Atg11, which would require avidity for a stable interaction of both proteins. We have included this data as Fig. 3b in the revised manuscript.

Figure 3b

It should also be noted that also GFP- μ NS, without the Atg11 fusion, has been published to form particles with different sizes in the cytoplasm of yeast cells (Munder et al., Elife 2016, Video1), however it is unclear if these are aggregates or ordered oligomers (Broehring et al., 2002, McCutcheon et al., Virology 1999). We quantified the size of μ NS particles. Whereas GFP- μ NS particles show in average a slightly smaller size than Atg11- μ NS fusions, part of the Atg11- μ NS particles have a similar size to that of μ NS (see Supplementary Fig. 1b). Moreover, these smaller Atg11- μ NS particles are sufficient to promote Vac8 and vacuolar binding, supporting that vacuolar binding of Atg11- μ NS does not stem from an indirect effect due to large aggregate formation (Reviewer Figure R5).

Reviewer Figure R5: Atg11-GFP- μ NS particles of various size are recruited to the vacuole and are able to induce membrane deformation.

4) If Vac8 is clustered by its recruitment to Ape1 complexes and Atg11 by its fusion to μ NS, both puncta colocalize and the authors presented this as a proof of avidity (Figure 3 F and G). In this scenario, one would expect to see two puncta ‘touching’ each other. However, the authors observed that GFP fluorescence was entirely surrounded and embedded by BFP. An alternative interpretation of the data would be that Vac8 recruits Atg11 independently of clustering and that clustering of Atg11 follows initial recruitment. Although the authors did a control experiment using GFP-Atg11 (not tagged with μ NS), the two experiments are different. In Figure 3D Atg11 is tagged at its N-term whereas in Figure 3F it is C-terminally tagged. The authors need to repeat this critical experiment with only one type of fusion protein to exclude that tagging impairs protein function.

Furthermore, it might be difficult to reveal GFP-Atg11 fluorescence in the absence of clustering on *ot*-Vac8 Δ N dots due to lower abundance and weaker signal. The authors should use biochemical assays to prove that clustering of both proteins is required for efficient interaction. This is particularly important since an interaction of both proteins was observed using co-IP of PA-Atg11 with GFP-Vac8 (Figure 2G).

As requested by the reviewer, we have repeated this experiment also with C-terminally tagged Atg11-GFP. Similar to our previous findings, Atg11-GFP was unable to bind to ot-Vac8 Δ N-BFP. However, in the presence of the cargo receptor Atg19 Atg11-GFP was proficient in binding to BFP-Ape1 and clearly visible in our fluorescence microscopy setup, excluding that a possible lack of visibility, due to low abundance and/or a weaker signal, could lead to a misinterpretation (new Supplementary Fig. 4d and 4e). As described in point 3, we have also performed additional experiments to strengthen the avidity model (Fig. 3b).

d**e**
Supplementary Figure 4d and 4e

"Figure 3 F and G). In this scenario, one would expect to see two puncta 'touching' each other."

The puncta in Fig. 3f and 3g are actually touching and not overlapping. We apologize that the selected example and the colors used in the overlay made it difficult to see the exact position of the two puncta. We have now selected a better example with less bright puncta, which makes the exact position easier to see (Fig. 3g), and provide additional images here (Reviewer Figure R6). We have also adjusted the text in the manuscript accordingly.

g
Figure 3g

Reviewer Figure R6: Representative images of *ot-Vac8ΔN-GFP* being in close association with *Atg11-BFP-μNS* particles.

5) The assembly of the PAS involves an ordered recruitment of the Atg protein machinery. To test, whether *Vac8* is important for initiation, elongation/maturation or fusion, the authors tested at which stage the recruitment-cascade was disrupted if *Vac8* was deleted. They found that puncta formation of *Atg8* and *Atg14* was reduced in *vac8Δ* cells, while *Atg9* was still recruited to *Ape1*. The authors further found that the PI3K complex was recruited by *Vac8* to the vacuolar membrane independently of its function in autophagy. It thus remains unclear whether the reduction in *Atg14* puncta formation observed in *vac8Δ* cells is related to autophagy or to the autophagy independent interaction of both. If this alternative interpretation is true, puncta formation of *Atg14* is only a consequence of *Vac8* clustering, which is consistent with data in Figure 5A, B. It remains thus unclear at which stage autophagy is blocked in *vac8Δ* cells and the conclusions that it is involved in maturation is not supported by data.

It is important to distinguish between

- 1. vacuole targeting of the PI3K, which depends on *Vac8* but not on autophagy, and
- 2. PI3K association with the PAS, which depends on its prior localization to the vacuole, but then furthermore requires *Atg9* at the PAS. These are two different events that happen sequentially and are both required for eventual selective autophagy function.

1. *Vacuolar recruitment of the PI3K complex by Vac8 is direct and does not require Atg11 (Fig. 4e and 4f). To strengthen this notion, we used the multiple knock-out strain created by the Klionsky lab, which lacks all core Atg proteins (including Atg11, Atg1 and Atg13), but Vac8 is present. In this strain, expression of Atg14-GFP together with its interaction partner Vps30 (also called Atg6) resulted in vacuolar localization, further supporting that core autophagy proteins are dispensable for the Vac8-dependent recruitment of the PI3K to the vacuole (new Supplementary Fig. 6d). (Note that Atg14 needs to be co-expressed with Vps30, as the absence of Vps30 also destabilizes Atg14, Kihara et al., JCB 2001)*

Supplementary Figure 6d

To further show that *Atg14* puncta formation is not a consequence of *Vac8* clustering, we have performed several experiments:

A. *Vac8* is known to enrich at vacuolar subdomains independently of autophagy, for instance at NVJs (Pan et al., MBoC 2000). Co-expression of *Vac8-mScarlet* and *Atg14-GFP* clearly shows that these two proteins do not enrich at the same sites on the vacuole (Reviewer Figure R7). If *Atg14* puncta formation was a consequence of *Vac8* clustering, then one would expect *Atg14-GFP* to also be enriched in the *Vac8*-enriched areas, which is not the case.

B. In the absence of *Atg11*, in which no cargo is recruited to the vacuole, *Atg14-GFP* also fails to form PAS puncta (Reviewer Figure R7). Nevertheless, *Atg14-GFP* is recruited to the vacuole and shows a mostly homogenous distribution (see also Fig. 4e and Supplementary Fig. 5f).

C. *Atg14* puncta formation requires the cargo and *Atg11*. But even when the *Atg11*-cargo complex is recruited to the vacuole, the *Atg14* puncta do not form when *Atg9* is absent (Fig. 4c and Supplementary Fig. 5b). This *Atg9* dependence furthermore proves that *Atg14* puncta formation cannot be a simple consequence of *Vac8* accumulating at the *Atg11*-cargo interaction site, as *Atg11-Vac8* clustering still takes place in *atg9Δ* cells. In addition to our data, this *Atg9*-dependency has been well established already before (He et al., MBoC 2008). Thus, PAS maturation beyond *Atg9* requires *Vac8*, due to its ability to recruit the PI3K to the vacuole, only then the PI3K can interact with *Atg9* vesicles (or *Atg9* associated components) to associate with the PAS.

Reviewer Figure R7: Atg14-GFP does not preferentially localize to Vac8-enriched regions in *atg11Δ* cells.

6) In Figure 5C, it is shown that targeting of Atg14 to the vacuole is sufficient to recruit Atg11/Ape1 complexes in the absence of Vac8. Why do the authors conclude that both, tethering of Atg11 and Atg14 at the vacuole is **required for PAS formation**, if tethering of Atg14 seems to be sufficient as indicated by the data. Altogether, it appears that **clustering of Vac8 at the vacuolar membrane recruits Atg14 which in turn recruits Atg11**. The argument that Atg11 induces clustering of Vac8 (Figure 1D and S2D) is not very convincing.

Vac8 microdomains at the vacuole are formed also independently of autophagy (e.g. NVJs), but do not enrich Atg14 per se. This has also been reported in the literature (e.g. Munzel et al., Autophagy 2020) and we have included further data supporting this (see point 5 above).

However, the Atg11-cargo complex (or Atg11-μNS) can induce the accumulation of Vac8 at the vacuolar membrane, where the Atg11-cargo complex and the vacuole touch. Atg11-μNS demonstrates that Vac8 is mobile and a large number of interaction sites (Atg11) can result in an extreme enrichment of Vac8. In wild type cells the Ape1 cargo is rather small and much fewer Atg11 are present, thus the enrichment of Vac8 at the vacuolar membrane is less well visible. Atg11-cargo recruitment to the vacuole takes place efficiently in the absence of Atg14 and Atg9 (Fig. 1a). Also, Atg9 is recruited to the Atg11-cargo complex in the absence of Atg14, demonstrating that the Atg11-Vac8 interaction is sufficient to initiate PAS formation at the vacuole, independently of Atg14. In contrast, Atg14 PAS recruitment requires Atg9 and the cargo complex (Fig. 4c). This is in line with the well-established hierarchy of Atg protein recruitment to the PAS, which is illustrated in Fig. 4a, and reported in the literature (see Hollenstein et al., Curr Op. 2020 for a review on the relevant literature on this). Tethering of Atg14 to the vacuole likely allows the co-recruitment of Atg9, and therefore Atg11, however, PAS formation is less efficient in this setup. Our data clearly shows that the initial recruitment of the Atg11-cargo complex does not require Atg14 (Fig. 1a). However, we agree with the reviewer that the interaction between Atg14 and

the PAS might contribute to stabilize the vacuole anchoring of the PAS. This could play a more important role at later steps during phagophore formation, which we do not address in this study. We discuss this possibility in the manuscript:

„Although Atg14 was dispensable for the initial recruitment of the Atg11-cargo complex to the vacuolar membrane, the PI3KC1 might play a role in stabilizing the PAS-vacuole connection during phagophore formation (Fig. 5c and 5d, column 4)“

7) The authors reconstituted the assembly of the PAS ectopically in vivo by targeting Vac8 to the nucleus. In Figure 6B, Vac8 clusters independently of Atg11 cargo complexes on nuclear membranes, while at Vac8-Atg11 contact sites no clustering of Vac8 was observed. This suggests that the approach does not fully recapitulate PAS assembly.

The enrichment of Vac8 at the native PAS (Ape1-cargo complex) is also not easily detectable, due to the small size of the native cargo and the high overall levels of Vac8 at the vacuole. A similar situation is present when Vac8 is tethered to the nucleus: the native Ape1-cargo complex is small in size and the nuclear Vac8 signal quite strong. Therefore, also here, as in the wild type situation, it is hard to see Vac8 enrichment at the nuclear PAS. Nevertheless, this enrichment is visible, for instance in Supplementary Fig. 8a, which we now point out with an arrow. Further examples are shown in Reviewer Figure R8.

Reviewer Figure R8: Representative examples of Vac8-mScarlet enrichment at the nuclear PAS.

Due to this difficulty in visualization, we have used Atg11- μ NS to demonstrate an enrichment of vacuolar Vac8 at the Atg11 contact site (Fig. 1f). We have now also repeated this for the nuclear tethered Vac8 (Supplementary Fig. 8b), which makes the enrichment clearly visible. Thus, both vacuolar and nuclear Vac8 do enrich at the contact site of Atg11-particles, the size of Atg11 particles, however, determines how well one can see this enrichment of Vac8.

Supplementary Figure 8b

Furthermore, the colocalization of Vps15 with nucleus targeted Vac8 is not convincing. The authors need to analyze protein recruitment by analyzing protein levels in nuclear membrane factions by western-blotting.

The nuclear signal of Vps15 was clear and has been quantified in a double blind manner (Fig. 6c). We thank the reviewer for pointing out that we have chosen a non-ideal example and have replaced the representative image in Fig. 6c and Supplementary Fig. 8c. In addition, we show here several additional representative images (Reviewer Figure R9).

Reviewer Figure R9: Representative images of Vps15-GFP localization in wild type and *vac8Δ* cells.

Interestingly, Atg2 puncta colocalize with Vac8 clusters at the nucleus, suggesting that PAS assembly does not depend on Atg11 clustering. The authors should analyze recruitment of Atg2 by nt-Vac8 in *atg11Δ* cells.

As explained in points 6 and 7, autophagy-specific Vac8 enrichment at the membrane is induced by cargo recruitment and requires the interaction of clustered Atg11 with

membrane-tethered Vac8. Other Vac8 enrichments on membranes are independent of autophagy.

It has been established that the PAS association of Atg2-Atg18 depends on the ability of Atg2 to bind Atg9 and PI3P (Obara JBC 2008, Kobayashi et al., FEBS Letters 2012, Gómez-Sánchez. et. al, JCB 2018). It has also been shown that PI3K recruitment to the PAS, and therefore also Atg2 recruitment, depends on Atg9 (He et al., MBoC 2008). Atg2 PAS association therefore requires all upstream proteins including Atg14, as well as PI3P production by the PI3K.

To strengthen the point that the same hierarchy and PI3P dependence also exists at the nuclear PAS, we have performed the experiment suggested by the reviewer. Indeed, deletion of Atg11 abrogates the formation also of nuclear Atg2 puncta (Supplementary Fig. 8e and 8f). We have explained this now also better in the revised text:

“Functional PI3KC1 produces PI3P at the PAS and thereby allows the recruitment of the PI3P dependent autophagy factors, the Atg2-Atg18 complex and Atg21 (Fig. 4a). To test if PI3P production at this ectopic PAS is functional, we monitored the formation of Atg2 puncta, which is a well-established readout for the production of PI3P at the PAS and the subsequent recruitment of the Atg2-Atg18 complex (Obara et al. JBC 2008, Kobayashi et al. FEBS 2012). Strikingly, Atg2-GFP formed puncta in *vac8Δ atg8Δ* cells expressing nt-Vac8ΔN-mScarlet, but not in *vac8Δ atg14Δ* cells, *vac8Δ atg11Δ* cells or those expressing nt-mScarlet, suggesting that Atg2-GFP associates with the ectopic PAS in a PI3P dependent manner (Fig. 6d and Supplementary Figure 7c).“

Supplementary Figure 8e and 8f

8) The recruitment of Atg2 to the “nuclear PAS” appears to depend on Atg14 and the authors concluded that PI3P production at the vacuole is required for this process. However, an alternative interpretation could be that clustering of Vac8 leads to an increase in local Atg2 which is easier to detect (Figure S7C). If the authors want to maintain the conclusion that PI3P is produced at the nuclear membrane, a direct assay would be required (e.g. PI3P sensor colocalization).

We cannot exclude that a low amount of Atg2 is recruited to the vacuole, or to the nuclear membrane in the nucleus tethered nt-Vac8 situation, which would not be detected by fluorescence microscopy due to its low abundance. This pool of Atg2 however would not be sufficient for autophagy function, as the PI3P dependent recruitment of the Atg2-Atg18 complex has been characterized in detail to be required for autophagy function (see point 7 above).

Moreover, non-autophagic Vac8 enrichments at the vacuole or the nucleus in the nt-Vac8 situation do not result in Atg2 recruitment (Supplementary Fig. 8e and Reviewer Figure R10), showing that Vac8 membrane enrichments are not sufficient to recruit Atg2.

Reviewer Figure R10: Atg2-GFP does not localize at nuclear Vac8 enrichments in the absence of nuclear PAS formation.

Also, it has been well established that Atg2-recruitment serves as a readout for PI3P production at the PAS, as PI3P production at the PAS is required for recruitment of Atg2 (see point 7 for more details). To further substantiate that the nuclear Atg2-GFP puncta are the result of Atg2 being recruited to a nuclear PAS, which forms similar to the vacuolar native PAS, we performed additional experiments. We tested whether the ectopic Atg2-GFP puncta formation was dependent on Atg11 and thus selective autophagy (Supplementary Fig. 8e). Second, we directly monitored whether the Atg2-GFP puncta overlap with BFP tagged Ape1, and hence co-localize with the selective autophagy cargo. We found that the formation of ectopic Atg2-GFP puncta was dependent on Atg11 and we observed that Atg2-GFP puncta co-localize with BFP-Ape1, demonstrating that these Atg2-GFP puncta indeed represent the PAS (Supplementary Fig. 8f). These findings further demonstrate that tethering of Vac8 to the nuclear membrane promotes PAS formation in the vicinity of the nucleus, and that PAS formation in the vicinity of the nucleus is similar to the natural PAS formation in the vicinity to the vacuole, i.e. it follows the same requirements.

We did not propose that PI3P is produced at the vacuolar or nuclear membrane, it is produced at the PAS, which is formed in vicinity to - but not continuous with - the vacuolar or nuclear membrane. We clarified this in the revised text:

" ... The production of PI3P on the phagophore thus might be spatially restricted to the region on the phagophore that is tethered to the vacuole, and then PI3P is distributed to the entire phagophore. "

9) The observation that Ape1 is transported to the vacuole in vac8Δ cells expressing nt-Vac8 (Figure 7A) could also rely on tight nuclear-vacuolar contact sites as present in Figure 7A. The conclusion that PAS initiation at the nucleus leads to the formation

of functional autophagosomes that deliver their content to the vacuole is thus not supported by this Figure. This conclusion would be significant and important, but more convincing data are needed. E.g. blocking fusion of autophagosomes with the vacuole will lead to an accumulation of Atg8-positive Ape1 puncta in *vac8Δ* cells expressing nt-Vac8 but not in cells lacking Vac8.

It has been well established in the literature that in the absence of Vac8 on the vacuole, no nucleus-vacuole junctions can form (Jeong et al., 2017; Kvam and Goldfarb, 2006; Pan et al., 2000). Therefore, Ape1 transport depending on the reconstituted nuclear PAS cannot involve such junctions. Ape1 transport to the vacuole has been well characterized, no alternative transport pathway than the Cvt pathway (or bulk autophagy under starvation conditions) has been observed. Atg19, the receptor for the Cvt pathway, is essential for Ape1 transport.

*To further strengthen the conclusion that PAS initiation at the nucleus leads to the formation of functional autophagosomes that deliver their content to the vacuole via the Cvt pathway, we added to Fig. 7a also exemplary images of an *atg19Δ* control. As expected for a delivery by the Cvt pathway, vacuolar uptake of BFP-Ape1 depended on Atg19.*

Figure 7a

In addition, we monitored GFP-Atg8 puncta formation in fusion deficient and fusion proficient nt-Vac8 cells. GFP-Atg8 puncta formation, which represent forming or mature autophagosomes, was enhanced in fusion-deficient nt-Vac8 cells, but not control cells, further strengthening our conclusions (Supplementary Fig. 9b and 9c). Together, these experiments clearly show that functional autophagosomes can be formed by a nuclear PAS, and that the delivery of the autophagosomal content to the vacuolar lumen depends on the CVT receptor Atg19.

Supplementary Figure 9b and 9c

Reviewers' Comments:

Reviewer #1:

Remarks to the Author:

The manuscript entitled "Spatial control of avidity regulates initiation and progression of selective autophagy." has been submitted as a revised original article by the Kraft group.

The authors have succeeded to address my critical points of the first review round and were able to convince me of the quality of their data.

Reviewer #2:

Remarks to the Author:

In the revised manuscript entitled "Spatiotemporal control of avidity regulates initiation and progression of selective autophagy" by Hollenstein et al., the authors have addressed all my queries satisfactorily. Now, I find the manuscript to be sufficient for publication in Nature Communications. I still have some minor comments.

1. There are a few typographical mistakes as follows:

Lines 300 and 329: "PI3KC1 complex" should be modified to PI3KC1. Please check mistakes again throughout the text.

Reviewer #3:

Remarks to the Author:

The authors have addressed all concerns by providing additional data or by detailed explanations. I therefore recommend the publication of this manuscript in its current form in Nat. commun.

Response to the points raised by the referees, 2

Reviewer #1 (Remarks to the Author):

The manuscript entitled "Spatial control of avidity regulates initiation and progression of selective autophagy." has been submitted as a revised original article by the Kraft group.

The authors have succeeded to address my critical points of the first review round and were able to convince me of the quality of their data.

Reviewer #2 (Remarks to the Author):

In the revised manuscript entitled "Spatiotemporal control of avidity regulates initiation and progression of selective autophagy" by Hollenstein et al., the authors have addressed all my queries satisfactorily. Now, I find the manuscript to be sufficient for publication in Nature Communications. I still have some minor comments.

1. There are a few typographical mistakes as follows:

Lines 300 and 329: "PI3KC1 complex" should be modified to PI3KC1. Please check mistakes again throughout the text.

We thank the reviewer for pointing out these mistakes and have corrected them.

Reviewer #3 (Remarks to the Author):

The authors have addressed all concerns by providing additional data or by detailed explanations. I therefore recommend the publication of this manuscript in its current form in Nat. communs.

We thank all three referees for their valuable input which helped improving our manuscript.